# Electrode pooling can boost the yield of extracellular recordings with switchable silicon probes

Kyu Hyun Lee[1,4], Yu-Li Ni [1,4], Jennifer Colonell[2], Bill Karsh[2], Jan Putzeys [3], Marius Pachitariu [2], Timothy D. Harris [2] & Markus Meister [1✉]

State-of-the-art silicon probes for electrical recording from neurons have thousands of recording sites. However, due to volume limitations there are typically many fewer wires carrying signals off the probe, which restricts the number of channels that can be recorded simultaneously. To overcome this fundamental constraint, we propose a method called electrode pooling that uses a single wire to serve many recording sites through a set of controllable switches. Here we present the framework behind this method and an experimental strategy to support it. We then demonstrate its feasibility by implementing electrode pooling on the Neuropixels 1.0 electrode array and characterizing its effect on signal and noise. Finally we use simulations to explore the conditions under which electrode pooling saves wires without compromising the content of the recordings. We make recommendations on the design of future devices to take advantage of this strategy.

[1] Division of Biology and Biological Engineering, Caltech, Pasadena, CA, USA. [2] HHMI Janelia Research Campus, Ashburn, VA, USA. [3] IMEC, Leuven, Belgium. [4] These authors contributed equally: Kyu Hyun Lee, Yu-Li Ni. ✉email: meister@caltech.edu

Understanding brain function requires monitoring the complex pattern of activity distributed across many neuronal circuits. To this end, the BRAIN Initiative has called for the development of technologies for recording "dynamic neuronal activity from complete neural networks, over long periods, in all areas of the brain", ideally "monitoring all neurons in a circuit"[1]. Recent advances in the design and manufacturing of silicon-based neural probes have answered this challenge with new devices that have thousands of recording sites[2–6]. Still, the best methods sample neural circuits very sparsely[7], for example recording fewer than $10^4$ cells in a mouse brain that has $10^8$.

In many of these electrode array devices only a small fraction of the recording sites can be used at once. The reason is that neural signals must be brought out of the brain via wires, which take up much more volume than the recording sites themselves. For example, in one state-of-the-art silicon shank, each wire displaces thirty times more volume than a recording site once the shank is fully inserted in the brain[2]. The current silicon arrays invariably displace more neurons than they record, and thus the goal of "monitoring all neurons" seems unattainable by simply scaling the present approach (but see ref. [8]). Clearly we need a way to increase the number of neurons recorded while avoiding a concomitant increase in the number of wires that enter the brain.

A common approach by which a single wire can convey multiple analog signals is time-division multiplexing[9]. A rapid switch cycles through the N input signals and connects each input to the output line for a brief interval (Fig. 1a). At the other end of the line, a synchronized switch demultiplexes the N signals again. In this way, a single wire carries signals from all its associated electrodes interleaved in time. The cycling rate of the switch is constrained by the sampling theorem[10]: It should be at least twice the highest frequency component present in the signal. The raw voltage signals from extracellular electrodes include thermal noise that extends far into the Megahertz regime. Therefore an essential element of any such multiplexing scheme is an analog low-pass filter associated with each electrode. This anti-alias filter removes the high-frequency noise above a certain cut-off frequency. In practice, the cut-off is chosen to match the bandwidth of neuronal action potentials, typically 10 kHz. Then the multiplexer switch can safely cycle at a few times that cut-off frequency.

Given the ubiquity of time-division multiplexing in communication electronics, what prevents its use for neural recording devices? One obstacle is the physical size of the anti-alias filter associated with each electrode. When implemented in CMOS technology, such a low-pass filter occupies an area much larger than the recording site itself[11], which would force the electrodes apart and prevent any high-density recording. What if one simply omitted the low-pass filter? In that case aliasing of high-frequency thermal fluctuations will increase the noise power in the recording by a factor equal to the number of electrodes N being multiplexed. One such device with a multiplexing factor of $N = 128$ has indeed proven unsuitable for recording action potentials, as the noise drowns out any signal[12]. A recent design with a more modest $N = 8$ still produces noise power 4–15 times higher than in comparable systems without multiplexing[13].

Other issues further limit the use of time-division multiplexing: The requirement for amplification, filtering, and rapid switching right next to the recording site means that electric power gets dissipated on location, heating up exactly the neurons one wants to monitor. Furthermore, the active electronics in the local amplifier are sensitive to light, which can produce artifacts during bright light flashes for optogenetic stimulation[2,14].

An alternative approach involves static electrode selection (Fig. 1b). Again, there is an electronic switch that connects the wire to one of many electrodes. However, the switch setting remains unchanged during the electrical recording. In this way, the low-pass filtering and amplification can occur at the other end of the wire, outside the brain, where space is less constrained. The switch itself requires only minimal circuitry that fits comfortably under each recording site, even at a pitch of 20 μm or less. Because there is no local amplification or dynamic switching, the issues of heat dissipation or photosensitivity do not arise. This method has been incorporated recently into flat electrode arrays[15] as well as silicon prongs[2,6,16]. It allows the user to choose one of many electrodes intelligently, for example, because it carries a strong signal from a neuron of interest. This strategy can increase the yield of neural recordings, but it does not increase the number of neurons per wire.

On this background, we introduce a third method of mapping electrodes to wires: select multiple electrodes with suitable signals and connect them to the same wire (Fig. 1c). Instead of rapidly cycling the intervening switches, as in multiplexing, simply leave all those switches closed. This creates a "pool" of electrodes whose signals are averaged and transmitted on the same wire. At first, that approach seems counterproductive, as it mixes together recordings that one would like to analyze separately. How can one ever reconstruct which neural signal came from which electrode? Existing multi-electrode systems avoid this signal mixing at all cost, often quoting the low cross-talk between channels as a figure of merit. Instead, we will show that the pooled signal can be unmixed if one controls the switch settings carefully during the recording session. Under suitable conditions, this method can record many neurons per wire without appreciable loss of information.

We emphasize that the ideal electrode array device to implement this recording method does not yet exist. It would be entirely within reach of current fabrication capabilities, but every new silicon probe design requires a substantial investment and consideration of various trade-offs. With this article, we hope to make the community of electrode users aware of the opportunities in this domain and start a discussion about future array designs that use intelligent electrode switching, adapted to various applications in basic and translational neuroscience.

## Theory

**Motivation for electrode pooling: spike trains are sparse in time.** A typical neuron may fire ~10 spikes/s on average[17]. Each action potential lasts for ~1 ms. Therefore this neuron's signal occupies <1% of the time axis in an extracellular recording (e.g., Fig. 3b). Sometimes additional neurons lie close enough to the

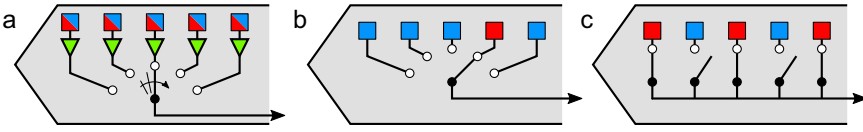

**Fig. 1 Strategies for using a single wire to serve many recording sites in switchable silicon probes. a** Time-division multiplexing. Rapidly cycling the selector switch allows a single wire to carry signals from many recording sites interleaved in time. Triangles represent anti-aliasing filters. **b** Static switching. A single wire connects to one of many possible recording sites through a selector switch. **c** Electrode pooling. Many recording sites are connected to a single wire through multiple controllable switches.

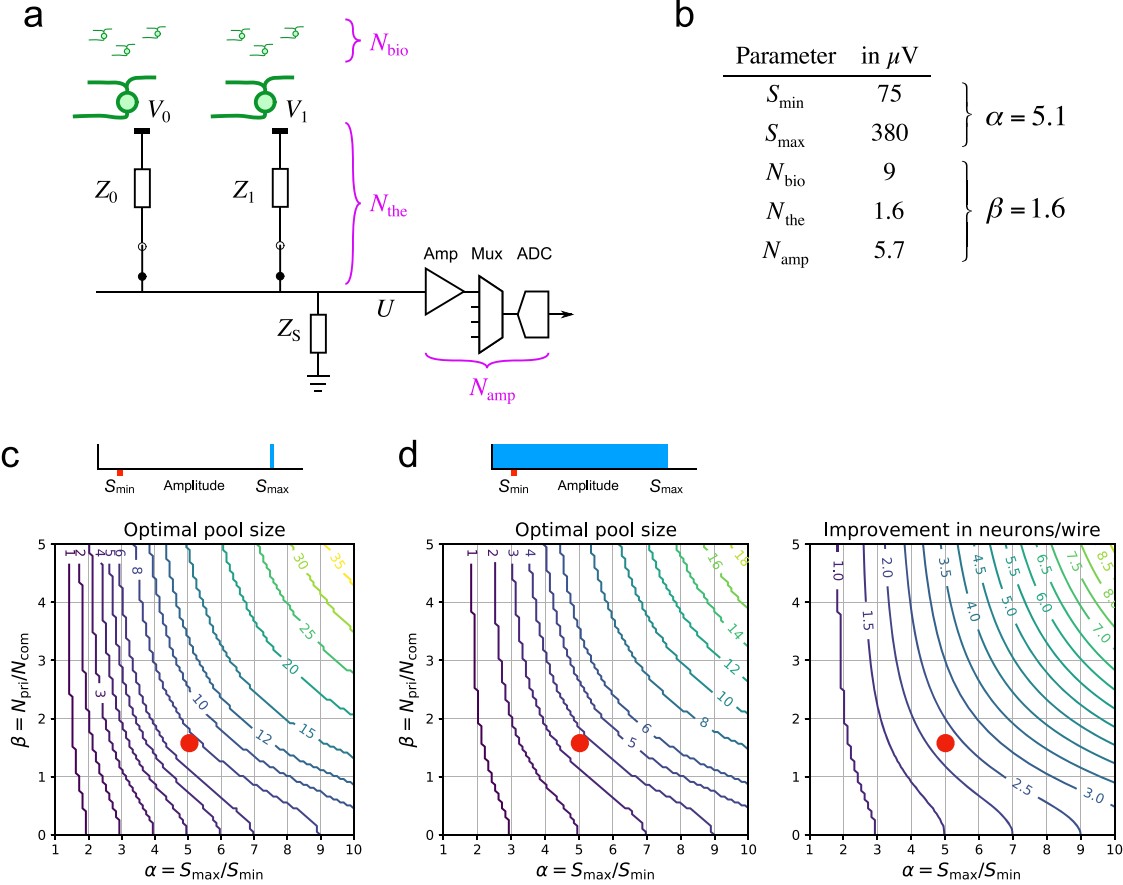

**Fig. 2 Pooling of signal and noise. a** An idealized circuit for two electrodes connected to a common wire along with downstream components of the signal chain, such as the amplifier, multiplexer, and digitizer. $Z_0$, $Z_1$: total impedance for electrodes 0 and 1, with contributions from the metal/bath interface and the external bath. $Z_S$: shunt impedance at the amplifier input. Noise sources include biological noise from distant neurons ($N_{bio}$); thermal noise from the electrode impedance ($N_{the}$), and common electronic noise from the amplifier and downstream components ($N_{amp}$). **b** Numerical values of the relevant parameters, derived from experiments or the literature (sections Experiments and Simulations). **c**, **d** The optimal electrode pool under different assumptions about the spike amplitude distribution (top insets). The contour plots show the optimal pool size and the enhancement of the neuron/wire ratio as a function of the parameters $\alpha$— the ratio of largest to smallest sortable spike signals—and $\beta$ —the ratio of private to common noise. **c** Most favorable condition: Each electrode carries a single large spike of amplitude $S_{max}$, and spikes are sortable down to amplitude $S_{min}$. In this case the neurons/ wire ratio is equal to the pool size. **d** Generic condition: Each electrode carries a uniform distribution of spike amplitudes between 0 and $S_{max}$. Red dots: Conditions of $\alpha$ and $\beta$ encountered experimentally, based on the values in panel b.

same electrode to produce large spikes. That still leaves most of the time axis unused for signal transmission. Electrode pooling gives the experimenter the freedom to add more neurons to that signal by choosing other electrodes that carry large spikes. Eventually a limit will be reached when the spikes of different neurons collide and overlap in time so they can no longer be distinguished. These overlaps may be more common under conditions where neurons are synchronized to each other or to external events.

**The effects of pooling on spikes and noise**. What signal actually results when one connects two electrodes to the same wire? Figure 2a shows an idealized circuit for a hypothetical electrode array that allows electrode pooling. Here the common wire is connected via programmable switches to two recording sites. At each site $i$, the extracellular signal of nearby neurons reaches the shared wire through a total electrode impedance $Z_i$. This impedance has contributions from the metal/saline interface and the external electrolyte bath[18,19], typically amounting to 100 kΩ–1 MΩ. By comparison, the CMOS switches have low impedance, typically ~100 Ω[18], which we will ignore.

In general, one must also consider the shunt impedance $Z_S$ in parallel to the amplifier input. This can result from current leaks along the long wires as well as the internal input impedance of the amplifier. For well-designed systems, this shunt impedance should be much larger than the electrode impedances; for the Neuropixels device, we will show that the ratio is at least 100. Thus one can safely ignore it for the purpose of the present approximations. In that case, the circuit acts as a simple voltage divider between the impedances $Z_i$. If a total of $M$ electrodes are connected to the shared wire, the output voltage $U$ is the average of the signals at the recording sites $V_i$, weighted inversely by the electrode impedances,

$$U = \sum_{i=1}^{M} c_i V_i \qquad (1)$$

where

$$c_i = \frac{1/Z_i}{\sum_{j=1}^{M} 1/Z_j} \qquad (2)$$

is defined as the pooling coefficient for electrode $i$. If all electrodes

have the same size and surface coating, they will have the same impedance, and in that limit one expects the simple relationship

$$U = \frac{1}{M} \sum_{i=1}^{M} V_i \qquad (3)$$

Thus an action potential that appears on only one of the $M$ electrodes will be attenuated in the pooled signal by a factor $\frac{1}{M}$.

In order to understand the trade-offs of this method, we must similarly account for the pooling of noise (Fig. 2a). There are three relevant sources of noise: (1) thermal ("Johnson") noise from the impedance of the electrode; (2) biological noise ("hash") from many distant neurons whose signals are too small to be resolved; (3) electronic noise resulting from the downstream acquisition system, including amplifier, multiplexer, and analog-to-digital converter. The thermal noise is private to each electrode, in the sense that it is statistically independent of the noise at another electrode. The biological noise is similar across neighboring electrodes that observe the same distant populations[20]. For widely separated electrodes the hash will be independent and thus private to each electrode, although details depend on the neuronal geometries and the degree of synchronization of distant neurons[21]. In that case the private noise is

$$N_{\text{pri},i} = \sqrt{N_{\text{the},i}^2 + N_{\text{bio},i}^2} \qquad (4)$$

because thermal noise and biological noise are additive and statistically independent.

Finally the noise introduced by the amplifier and data acquisition is common to all the electrodes that share the same wire,

$$N_{\text{com}} = N_{\text{amp}} \qquad (5)$$

In the course of pooling, the private noise gets attenuated by the pooling coefficient $c_i$ (Eq. (2)) and summed with contributions from other electrodes. Then the pooled private noise gets added to the common noise from data acquisition, which again is statistically independent of the other noise sources. With these assumptions the total noise at the output has RMS amplitude

$$N_{\text{tot}} = \sqrt{N_{\text{com}}^2 + \sum_{i=1}^{M} c_i^2 N_{\text{pri},i}^2} \qquad (6)$$

If all electrodes have similar noise properties and impedances this simplifies to

$$N_{\text{tot}} = \sqrt{N_{\text{com}}^2 + N_{\text{pri}}^2/M} \qquad (7)$$

**Theoretical benefits of pooling**. Now we are in a position to estimate the benefits from electrode pooling. Suppose that the electrode array records neurons with a range of spike amplitudes: from the largest, with spike amplitude $S_{\text{max}}$, to the smallest that can still be sorted reliably from the noise, with amplitude $S_{\text{min}}$. To create the most favorable conditions for pooling one would select electrodes that each carry a single neuron, with spike amplitude $\sim S_{\text{max}}$ (Fig. 2c). As one adds more of these electrodes to the pool, there comes a point when the pooled signal is so attenuated that the spikes are no longer sortable from the noise. Pooling is beneficial as long as the signal-to-noise ratio of spikes in the pooled signal is larger than that of the smallest sortable spikes in a single-site recording, namely

$$\frac{S_{\text{max}}/M}{\sqrt{N_{\text{com}}^2 + N_{\text{pri}}^2/M}} > \frac{S_{\text{min}}}{\sqrt{N_{\text{com}}^2 + N_{\text{pri}}^2}} \qquad (8)$$

This leads to a limit on the pool size $M$,

$$M < M_{\text{max}} = \sqrt{\left(\frac{\beta^2}{2}\right)^2 + (1 + \beta^2)\alpha^2} - \frac{\beta^2}{2} \qquad (9)$$

where

$$\alpha = S_{\text{max}}/S_{\text{min}}, \quad \beta = N_{\text{pri}}/N_{\text{com}} \qquad (10)$$

If one pools more than $M_{\text{max}}$ electrodes all the neurons drop below the threshold for sorting. So the optimal pool size $M_{\text{max}}$ is also the largest achievable number of neurons per wire. This number depends on two parameters: the ratio of private to common noise, and the ratio of largest to smallest useful spike amplitudes (Eq. (10)). These parameters vary across applications, because they depend on the target brain area, the recording hardware, and the spike-sorting software. In general, users can estimate the parameters $\alpha$ and $\beta$ from their own experience with conventional recordings, and find $M_{\text{max}}$ from the graph in Fig. 2c.

Next we consider a more generic situation, in which each electrode carries a range of spikes from different neurons (Fig. 2d). For simplicity, we assume a uniform distribution of spike amplitudes between 0 and $S_{\text{max}}$. As more electrodes are added to the pool, all the spikes are attenuated, so the smallest action potentials drop below the sortable threshold $S_{\text{min}}$. Beyond a certain optimal pool size, more spikes are lost in the noise than are added at the top of the distribution, and the total number of neurons decreases. By the same arguments used above one finds that the improvement in the number of sortable neurons, $n_M$, relative to conventional split recording, $n_1$, is

$$\frac{n_M}{n_1} = \frac{M\left(\alpha - M\sqrt{\frac{1 + \beta^2/M}{1 + \beta^2}}\right)}{\alpha - 1} \qquad (11)$$

The optimal pool size $M_{\text{max}}$ is the $M$ which maximizes that factor. The results are plotted in Fig. 2d.

The benefits of pooling are quite substantial if the user can select electrodes that carry large spikes. For example under conditions of $\alpha$ and $\beta$ that we have encountered in practice, Fig. 2c predicts that one can pool 8 electrodes and still resolve all the signals, thus increasing the neuron/wire ratio by a factor of 8. On the other extreme—with a uniform distribution of spike amplitudes—the optimal pool of 4 electrodes increases the neuron/wire ratio by a more modest but still respectable factor of 2.3 compared to conventional recording. The following section explains how one can maximize that yield.

**Acquisition and analysis of pooled recordings**. With these insights about the constraints posed by signal and noise one can propose an overall workflow for experiments using electrode pooling (Fig. 3a). A key requirement is that the experimenter can control the switches that map electrodes to wires. This map should be adjusted to the unpredictable contingencies of any particular neural recording experiment. In fact, the experimenter will benefit from using different switch settings during the same session.

A recording session begins with a short period of acquisition in "split mode" with only one electrode per wire. The purpose is to acquire samples of the spike waveforms from all neurons that might be recorded by the entire array. If the device has $E$ electrodes and $W$ wires, this sampling stage will require $E/W$ segments of recording to cover all electrodes. For each segment, the switches are reset to select a different batch of electrodes. Each batch should cover a local group of electrodes, ensuring that the entire "footprint" of each neuron can be captured.

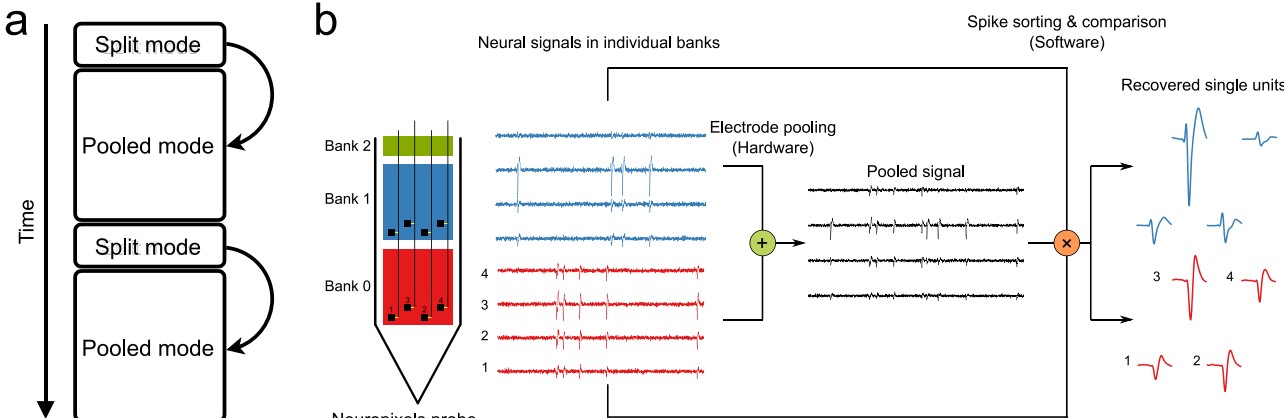

**Fig. 3 Workflow proposed for electrode pooling. a** Time line of an experiment, alternating short split, and long pooled recording sessions. **b** Electrode pooling using the Neuropixels probe. Recording sites (black squares, numbered from 1 to 4) in the same relative location of each bank can be pooled to a single wire by closing the switches (yellow). This generates the pooled signal (black), which is a weighted average of the signals detected in each bank (red and blue traces). From the pooled signal one recovers distinct spike shapes by spike-sorting. A comparison to the spike shapes observed in split-mode recordings allows the correct allocation of each spike to the electrodes of origin.

During this sampling stage, the experimenter performs a quick analysis to extract the relevant data that will inform the pooling process. In particular, this yields a catalog of single neurons that can be extracted by spike-sorting. For each of those neurons, one has the spike waveform observed on each electrode. Finally, for every electrode one measures the total noise. The amplifier noise $N_{amp}$ and thermal noise $N_{the}$ can be assessed ahead of time, because they are a property of the recording system, and from them, one obtains the biological noise $N_{bio}$. Now the experimenter has all the information needed to form useful electrode pools. Some principles one should consider in this process are:

1. Pool electrodes that carry large signals. Electrodes with smaller signals can contribute to smaller pools.
2. Pool electrodes with distinct spike waveforms.
3. Pool distant electrodes that don't share the same hash noise.
4. Don't pool electrodes that carry dense signals with high firing rates.

After allocating the available wires to effective electrode pools one begins the main recording session in pooled mode. Ideally this phase captures all neurons with spike signals that are within reach of the electrode array.

In analyzing these recordings the goal is to detect spikes in the pooled signals and assign each spike correctly to its electrode of origin. This can be achieved by using the split-mode recordings from the early sampling stage of the experiment. From the spike waveforms obtained in split-mode, one can predict how the corresponding spike appears in the pooled signals. Here it helps to know all the electrode impedances $Z_i$ so the weighted mix can be computed accurately (Eq. (1)). This prediction serves as a search template for spike-sorting the pooled recording.

By its very nature electrode pooling produces a dense neural signal with more instances of temporal overlap between spikes than the typical split-mode recording. This places special demands on the methods for spike detection and sorting. The conventional cluster-based algorithm (peak detection–temporal alignment–PCA–clustering) does not handle overlapping spikes well[22]. It assumes that the voltage signal is sparsely populated with rare events drawn from a small number of discrete waveforms. Two spikes that overlap in time produce an unusual waveform that cannot be categorized. Recently some methods have been developed that do not force these assumptions[23,24]. They explicitly model the recorded signal as an additive superposition of spikes and noise. The algorithm finds an efficient model that explains the signal by estimating both the spike waveform of each neuron and its associated set of spike times. These methods are well suited to the analysis of pooled recordings.

Because the spike templates are obtained from split-mode recordings at the beginning of the experiment, they are less affected by noise than if one had to identify them de novo from the pooled recordings. Nonetheless, it probably pays to monitor the development of spike shapes during the pooled recording. If they drift too much, for example, because the electrode array moves in the brain[25], then a recalibration by another split-mode session may be in order (Fig. 3a). Alternatively, electrode drift may be corrected in real time if signals from neighboring electrodes are available[6], a criterion that may flow into the selection of switches for pooling. Chronically implanted electrode arrays can record for months on end[6], and the library of spike shapes can be updated continuously and scanned for new pooling opportunities.

It should be clear that the proposed workflow relies heavily on automation by dedicated software. Of course, automation is already the rule with the large electrode arrays that include thousands of recording sites, and electrode pooling will require little more effort than conventional recording. Taking the newly announced Neuropixels 2.0 as a reference (5120 electrodes and 384 wires): Sampling for 5 min from each of the 13 groups of electrodes will take a bit over an hour. Spike-sorting of those signals will proceed in parallel with the sampling and require no additional time. Then the algorithm decides on the electrode pools, and the main recording session starts. Note that these same steps also apply in conventional recording: The user still has to choose 384 electrodes among the 5120 options, and will want to scan the whole array to see where the best signals are. The algorithms we advocate to steer electrode pooling will simply become part of the software suite that runs data acquisition.

## Experiments

**Pooling characteristics of the Neuropixels 1.0 array.** To test the biophysical assumptions underlying electrode pooling, we used the Neuropixels probe version 1.0[2,16]. This electrode array has a single silicon shank with 960 recording sites that can be connected to 384 wires via controllable switches (Fig. 3b). The electrodes are divided into three banks (called Bank 0, Bank 1, and Bank 2 from the tip to the base of the shank). In the present

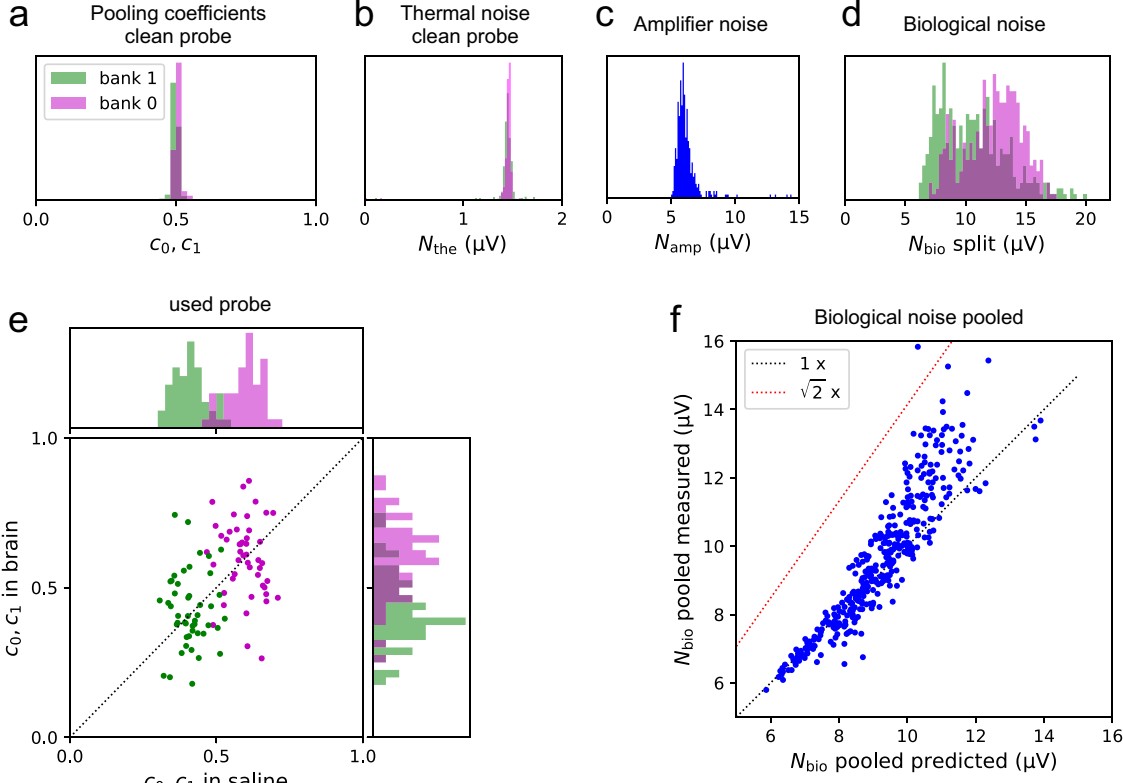

**Fig. 4 Pooling of signal and noise with the Neuropixels 1.0 device. a** Pooling coefficients on a pristine probe measured in saline, histogram across all sites in banks 0 (red) and 1 (green). **b** Thermal noise (RMS) during split recording in standard saline, histogram across all sites in banks 0 and 1. **c** Amplifier noise (RMS), histogram across all 383 wires. **d** Biological noise (RMS) during brain recordings, histogram across all sites in banks 0 and 1. **e** Pooling coefficients on a used probe, measured in saline (horizontal) vs in brain (vertical). 47 pairs of sites in banks 0 and 1 with suitable action potentials. **f** Biological noise in a pooled recording measured in brain (vertical) vs the prediction derived from assuming uncorrelated noise at the two sites. `1 x`: identity. `$\sqrt{2}$ x`: expectation for perfectly correlated noise.

study, only Banks 0 and 1 were used. Banks 0 and 1 each contain 383 recording sites (one channel is used for a reference signal). Each site has a dedicated switch by which it may connect to an adjacent wire. Sites at the same relative location in a bank share the same wire. These two electrodes are separated by 3.84 mm along the shank. Under the conventional operation of Neuropixels[2], each wire connects to only one site at a time. However, with modifications of the firmware on the device and the user interface we engineered independent control of all the switches. This enabled a limited version of electrode pooling across Banks 0 and 1.

We set out to measure those electronic properties of the device that affect the efficacy of pooling, specifically the split of the noise signal into common amplifier noise $N_{amp}$ (Eq. (7)) and private thermal noise $N_{the}$ (Eq. (4)), as well as the pooling coefficients $c_i$ (Eq. (2)). These parameters are not important for conventional recording, and thus are not quoted in the Neuropixels user manual, but they can be derived from measurements performed in a saline bath (see Methods).

On a pristine unused probe, the pooling coefficients $c_0$ and $c_1$ for almost all sites were close to 0.5 (Fig. 4a), as expected from the idealized circuit (Fig. 2a) if the electrode impedances are all equal (Eq. (2)). Correspondingly the thermal noise was almost identical on all electrodes, with an RMS value of $1.45 \pm 0.10\,\mu V$ (Fig. 4b). The amplifier noise $N_{amp}$ exceeded the thermal noise substantially, amounting to $5.7\,\mu V$ RMS on average, and more than $12\,\mu V$ for a few of the wires (Fig. 4c). Because this noise source is shared across electrodes on the same wire, it lowers $\beta$ in Eq. (9) and can significantly affect electrode pooling.

**Neural recording**. Based on this electronic characterization of the Neuropixels probe we proceeded to test electrode pooling in vivo. Recall that each bank of electrodes extends over 3.84 mm of the shank, and one needs to implant more than one bank into the brain to accomplish any electrode pooling. Clearly, the opportunities for pooling on this device are limited; nonetheless, it serves as a useful testing ground for the method.

In the pilot experiment analyzed here, the probe was inserted into the brain of a head-fixed, awake mouse to a depth of ~6 mm. This involved all of Bank 0 and roughly half of Bank 1, and covered numerous brain areas from the medial preoptic area at the bottom to the retrosplenial cortex at the top. Following the work flow proposed in Fig. 3, we then recorded for ~10 min each from Bank 0 and Bank 1 in split mode, followed by ~10 min of recording from both banks simultaneously in pooled mode.

**Unmixing a pooled recording**. As proposed above, one can unmix the pooled recording by matching its action potentials to the spike waveforms sampled separately on each of the two banks (Fig. 3b). Each of the three recordings (split Bank 0, split Bank 1, and pooled Banks 0 + 1) was spike-sorted to isolate single units. Then we paired each split-mode unit with the pooled-mode unit that had the most similar waveform, based on the cosine similarity of their waveform vectors (Eq. (16), Fig. 5b). In most cases, the match was unambiguous even when multiple units were present in the two banks with similar electrode footprints (Fig. 5a). The matching algorithm proceeded iteratively until the similarity score for the best match dropped below 0.9 (Fig. 5b).

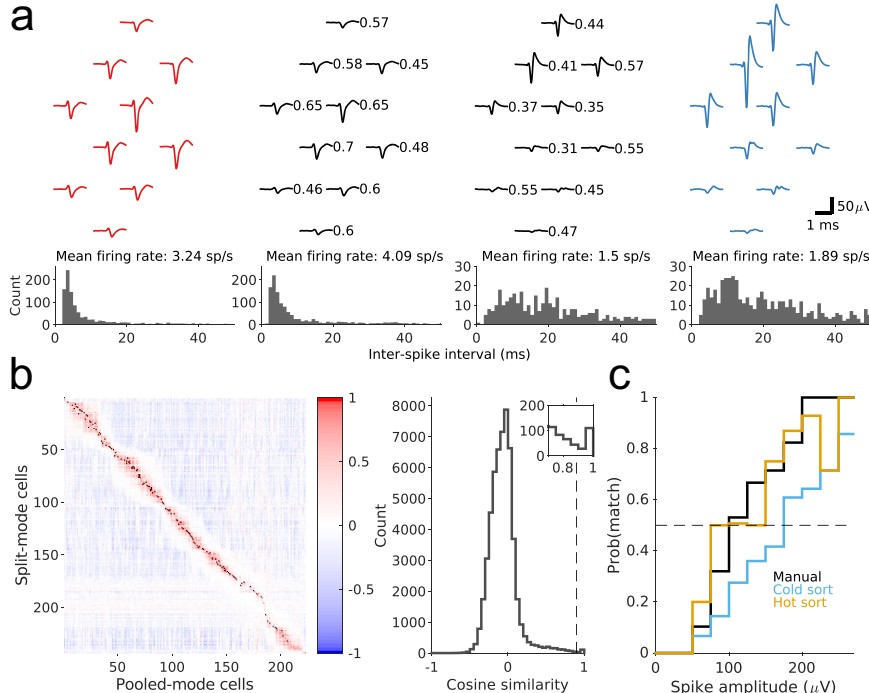

**Fig. 5 Recordings from mouse brain. a** Matching spike shapes from split- and pooled-mode recordings. Top: Waveforms of two sample units (middle, black) detected in pooled mode on the same set of wires. The left unit was matched to a unit recorded in split mode from Bank 0 (red) and the right unit to one from Bank 1 (blue). Numbers indicate the scaling of the signal of the pooled-mode unit relative to its split-mode signal. Bottom: the mean firing rates and the interspike-interval distributions are similar for the matched pairs. **b** Left: matrix of the cosine similarity between units recorded in pooled- and split-mode, arranged by depth. Black dot indicates greater than the threshold at 0.9. Right: distribution of the cosine similarity. Dashed line indicates the threshold at 0.9. Inset zooms into the 0.7–1 range of the distribution. **c** Fraction of units from the two split recordings that are matched to a unit in the pooled recording as a function of spike amplitude. Three different sorting conditions are shown: sorting all recordings by KiloSort1 followed by manual curation (Manual), sorting all recordings by KiloSort2 (Cold sort), and sorting the pooled recording by KiloSort2 with templates initialized from the split recordings (Hot sort). Dashed line indicates 50%, or the `break-even' point where the pooled-mode yields as many simultaneous recordings as the average split-mode.

We corroborated the resulting matches by comparing other statistics of the identified units, such as the mean firing rate and inter-spike-interval distribution (Fig. 5a).

When spike-sorting the pooled-mode recording there is of course a strong expectation for what the spike waveforms will be, namely a scaled version of spikes from the two split-mode recordings. This suggests that one might jump-start the sorting of the pooled signal by building in the prior knowledge from sorting the split-mode recordings. Any such regularization could be beneficial, not only to accelerate the process but to compensate for the lower SNR in the pooled signal. We explored this possibility by running the template-matching function of Kilo-Sort2 on the pooled-mode recording with templates from split-mode recordings ("hot sorting"). Then we compared this method to two other procedures (Fig. 5c): (1) sorting each recording separately, using KiloSort1 with manual curation ("manual"), and (2) sorting each recording separately using KiloSort2 with no manual intervention ("cold sorting").

Figure 5c illustrates what fraction of the units identified in both split mode recordings combined were recovered from the pooled recording, and how that fraction depends on the spike amplitude. First, this shows that hot sorting significantly outperforms cold sorting, and in fact rivals the performance of manually curated spike sorting. This is important, because manual sorting by a human operator will be unrealistic for the high-count electrode arrays in which electrode pooling may be applied. Second, one sees that the fraction of spikes recovered from the combined split recordings exceeds 0.5 even at moderate spike amplitudes of

100 µV. For spikes of that amplitude and above the pooled recording will contain more neurons than the average split recording. Clearly, electrode pooling is not restricted to the largest spikes in the distribution, but can be considered for moderate spike amplitudes as well.

Recall that the Neuropixels 1.0 probe is not optimized for electrode pooling, in that it has a fixed switching matrix, and only 2 banks of electrodes fit in the mouse brain. Thus our pilot experiments were limited to brute-force pooling the two banks site-for-site without regard to the design principles for electrode pools. Nonetheless, the "hot sorting" method recovered more neurons from the pooled recording (184) than on average over the two split recordings (166). We conservatively focused this assessment only on units identified in the split recordings, ignoring any unmatched units that appeared in the pooled recording. This validates the basic premise of electrode pooling even under the highly constrained conditions. Overall, the above sequence of operations demonstrates that a pooled-mode recording can be productively unmixed into the constituent signals, and the resulting units assigned to their locations along the multi-electrode shank.

**Pooling of signal and noise in vivo**. Closer analysis of the spike waveforms from split and pooled recordings allowed an assessment of the pooling coefficients in vivo. When spikes are present on the corresponding electrodes in both banks (as in Fig. 5a) one can measure the pooling coefficients $c_0$ and $c_1$ of Eq. (2).

Unexpectedly, instead of clustering near 0.5, these pooling coefficients varied over a wide range (Fig. 4e), at least by a factor of 3. The two banks had systematically different pooling coefficients, suggesting that the impedance was lower for electrodes near the tip of the array.

Following this in vivo recording we cleaned the electrode array by the recommended protocol (tergazyme/water) and then measured the pooling coefficients in saline. Again the pooling coefficients varied considerably across electrodes, although somewhat less than observed in vivo (Fig. 4e). Also the bath resistance of the electrodes was larger on average than on an unused probe (30 kΩ as opposed to 13 kΩ). This change may result from the interactions within brain tissue. For example, some material may bind to the electrode surface and thus raise its bath resistance. This would lower the pooling coefficient of the affected electrode and raise that of its partners. Because the thermal noise is never limiting (Fig. 4b–d), such a change would easily go unnoticed in conventional single-site recording. The precise reason for the use-dependent impedance remains to be understood.

To measure the contributions of biological noise in vivo we removed from the recorded traces all the detected spikes and analyzed the remaining waveforms. After subtracting (in quadrature) the known thermal and electrical noise at each site (Fig. 4b, c) one obtains the biological noise $N_{bio}$. This noise source substantially exceeded both the thermal and amplifier noise (Fig. 4f). It also showed different amplitude on the two banks, presumably owing to differences between brain areas 3.84 mm apart.

Given this large distance between electrodes in the two banks, one expects the biological noise to be statistically independent between the two sites, because neurons near one electrode will be out of reach of the other. To verify this in the present recordings we measured the biological noise in the pooled condition and compared the result to the prediction from the two split recordings, assuming that the noise was private to each site. Indeed the noise in the pooled signal was largely consistent with the assumption of independent noise (Fig. 4f). It seems likely that the 1-cm shank length on these and similar array devices suffices for finding electrodes that carry independent biological noise.

## Simulations

How many electrodes could experimenters pool and still sort every neuron with high accuracy? Earlier we had derived a theoretical limit to electrode pooling based solely on the signal and noise amplitudes (Fig. 2). To explore what additional limitations might arise based on the density of spikes in time and the needs of spike sorting we performed a limited simulation of the process (Fig. 6a). We simulated units with an extracellular footprint extending over 4 neighboring electrodes, and then pooled various such tetrodes into a single 4-channel recording. These pooled signals were then spike-sorted and the resulting spike trains compared to the known ground-truth spike times, applying a popular metric of accuracy[26]. This revealed how many neurons can be reliably recovered depending on the degree of electrode pooling (Fig. 6b). Then we evaluated the effects of various parameters of the simulation, such as the amplitude of the largest spikes, the biological noise, and the average firing rate.

For simplicity we focused on the favorable scenario of Fig. 2c: It presumes that the experimenter can choose for pooling a set of tetrodes that each carry a single unit plus noise. The curves of recovered units vs pool size have an inverted-U shape (Fig. 6b). For small electrode pools, one can reliably recover all the units. Eventually, however, some of the units drop out, and for a large pool size, all the recovered units fall below the desired quality

threshold. We will call the pool that produces the largest number of recovered units the "optimal pool".

For the "standard" condition of simulations, we chose a reasonably large spike amplitude of 380 μV peak-to-peak (the 90th percentile in a database of recordings by the Allen Institute[27]), a firing rate of 10 Hz, and all the noise values as determined experimentally from the Neuropixels 1.0 device (Fig. 4). Under these conditions, one can pool up to 5 electrodes per wire and still recover all 5 of the units reliably (Fig. 6b). This optimal pool size is sensitive to the amplitude of the spikes: If the spike amplitude is reduced by a factor of 2, the optimal pool drops from 5 to 3 electrodes. Similarly, if the biological noise increases to 15 μV, the optimal pool is reduced to 4 electrodes. This indicates that the recovery of the units from the pooled signal is strongly determined by the available signal-to-noise ratio at each electrode. By contrast, increasing the firing rate two-fold to 20 Hz did not change the optimal pool from 5. Thus the temporal overlap of spikes is not yet a serious constraint. Looking to the future, if the amplifier noise on each wire could be reduced by a factor of 2 the optimal pool would expand significantly from 5 to 7 electrodes or more (Fig. 6b).

How do these practical results relate to the theoretical bounds of Fig. 2? Recall that this bound depends on the noise properties, but also on the ratio of largest to smallest sortable spikes. In our "standard" simulation with a pool size of 1 (split mode) we found that the smallest sortable spikes had an amplitude of 75 μV. This also corresponds to the low end of sorted spikes reported by the Allen Institute (10th percentile[27]). With these bounds on large and small spikes, and the measured values of private and common noise, one obtains $\alpha = 5.1$ and $\beta = 1.6$ in Eq. (9), which predicts an optimal pool of $M_{max} = 8$ (Fig. 2c), compared to the actually observed value of 5. The simple theory based purely on signal and noise amplitude give a useful estimate, but additional practical constraints that arise from temporal processing and spike-sorting lower the yield somewhat from there.

In summary, under favorable conditions where the experimenter can select electrodes, the pooling method may increase the number of units recorded per wire by a factor of 5. Even for significantly smaller spikes or higher biological noise one can expect a factor of 3. And with future technical improvements a factor of 7 or more is plausible.

## Discussion

**Summary of results**. This work presents the concept of electrode pooling as a way to multiply the yield of large electrode arrays. We show how the signals from many recording sites can be combined onto a small number of wires, and then recovered by a combination of experimental strategy and spike-sorting software. The reduced requirement for wires coursing through the brain will lead to slender array devices that cause less damage to the neurons they are meant to observe. We developed the theory behind electrode pooling, analyzed the trade-offs of the approach, derived a mathematical limit to pooling, and developed a recipe for experiment and analysis that implements the procedure (Figs. 2, 3). We also verified the basic assumptions about signal mixing and unmixing using a real existing device: the Neuropixels 1.0 probe (Figs. 4, 5). We showed that signals from different neurons can be reliably disambiguated and assigned back to the electrodes of origin. For the optimal design of electrode pools and to analyze the resulting data, it is advantageous to gather precise information about the impedance and noise properties of the device. In simulations, we showed that with a proper selection of electrodes based on the signals they carry, the method could improve the yield of neurons per wire by a factor of 3–7 (Fig. 6).

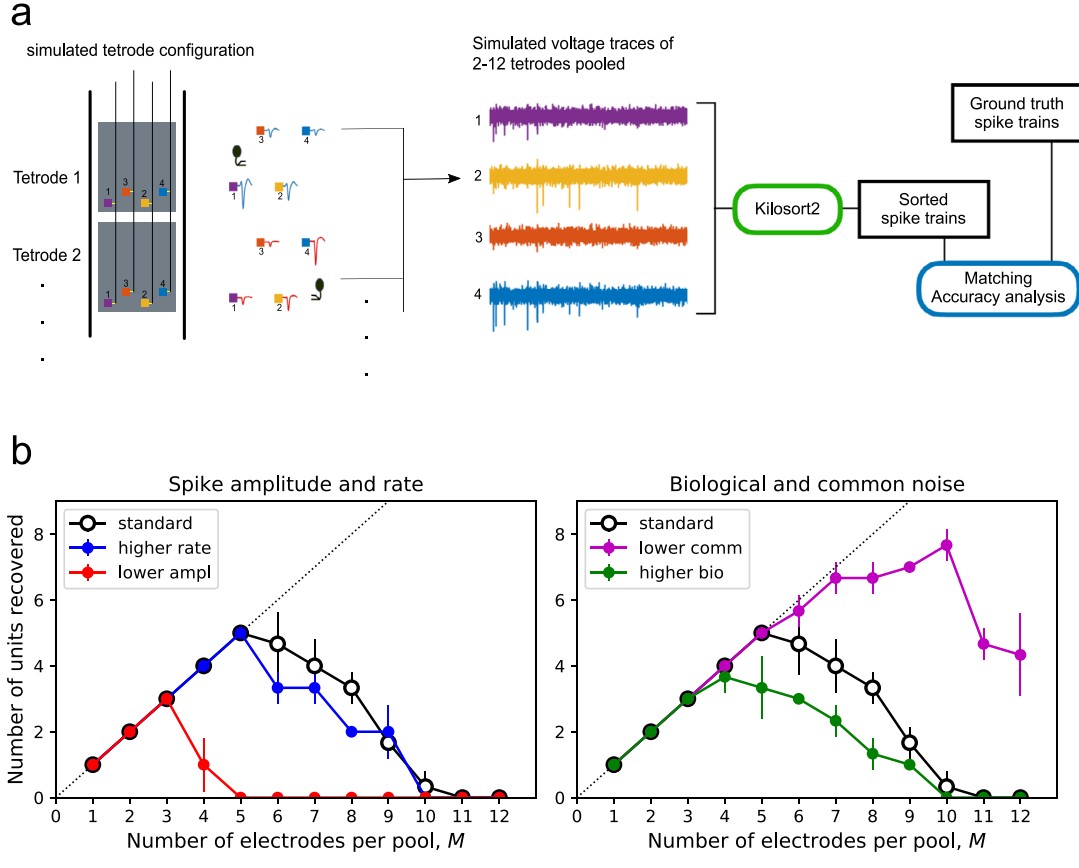

**Fig. 6 Simulations of electrode pooling. a** Workflow: Groups of four recording sites ("tetrodes") each carry a spike train from one simulated unit, superposed with electrode noise and biological noise. Between $M = 1$ and 12 of these tetrodes are then pooled into a single 4-wire recording followed by the addition of common noise. The pooled signal is sorted and the resulting single-unit spike trains are matched with the ground truth spike trains from the $M$ tetrodes. Units with an accuracy metric $> 0.8$ are counted as recovered successfully. **b** Number of units recovered as a function of the pool size, $M$, under various conditions of the simulation. Effects of varying different parameters. The "standard" condition serves as a reference: Spike amplitude $V = 380\,\mu V$, spike rate $r = 10\,Hz$, electrode noise $N_{ele} = 1.6\,\mu V$, common noise $N_{com} = 5.7\,\mu V$, biological noise $N_{bio} = 9\,\mu V$. "lower ampl": $V = 205\,\mu V$. "higher rate": $r = 20\,Hz$. "higher bio": $N_{bio} = 15\,\mu V$. "lower com": $N_{com} = 2.85\,\mu V$. Each parameter combination was simulated three times with noise and spike times resampled, error bars are mean ± SD.

Electrode pooling is categorically different from most data compression schemes that have been proposed for neural recording systems[28–30]. In many of those applications, the goal is to reduce the bit rate for data transmission out of the brain, for example using a wireless link. By contrast, electrode pooling seeks to minimize the number of electrode wires one needs to stick into the brain to sample the neural signals, thus minimizing biological damage to the system under study. By itself, that doesn't reduce the bit rate, although it produces denser time series. For the optimal wireless recording system, both objectives—lower wire volume at the input and lower data volume at the output —should be combined, and their implementations are fully independent.

### Future developments
*Hardware.* The ability to service multiple recording sites with a single wire opens the door for much larger electrode arrays that nevertheless maintain a slim form factor and don't require any onboard signal processing. On the commercially available Neuropixels 1.0 device[2] the ratio of electrodes to wires is only 2.5, and thus there is little practical benefit to be gained from electrode pooling. In most circumstances, the user can probably use static selection to pick 40% of the electrodes and still monitor every possible neuron. By contrast, the recently announced Neuropixels 2.0 array[6] has an electrode:wire ratio of 13.3. Another device, currently in engineering test, will have 4416 sites on a single

45 mm shank, with electrode:wire ratio of 11.5. For the Neuropixels technology, the number of sites can grow with shank count and shank length while channel count is limited by base area and trace crowding on the shank. These new probes already offer substantial opportunities to pool electrodes. Indeed, Steinmetz et al.[6] report an example of pooling two-electrode banks, although their approach to unmixing the signals differs from that advocated here.

The design of effective electrode pools requires some flexibility in how recording sites are connected to wires. In the current Neuropixels technology, each electrode has only one associated wire, which constrains the choice of electrode pools. The CMOS switch itself is small, but the local memory to store the switch state occupies some silicon space[31]. Nonetheless one can implement 3 switches per electrode even on a very tight pitch[32]. When arranged in a hierarchical network[15] these switches could effect a rich diversity of pooling schemes adapted to the specifics of any given experiment (Fig. 7). For example, one could route any one electrode among a group of four to any one of three wires with two 1:4 switches (Fig. 7c). This requires just 1 bit of storage per electrode, as in the current Neuropixels probe[2].

Another hardware design feature that could greatly increase the capacity for electrode pooling: An optional analog inverter at each electrode (Fig. 7d). This is a simple CMOS circuit that changes the sign of the waveform[33] depending on a local switch setting. If

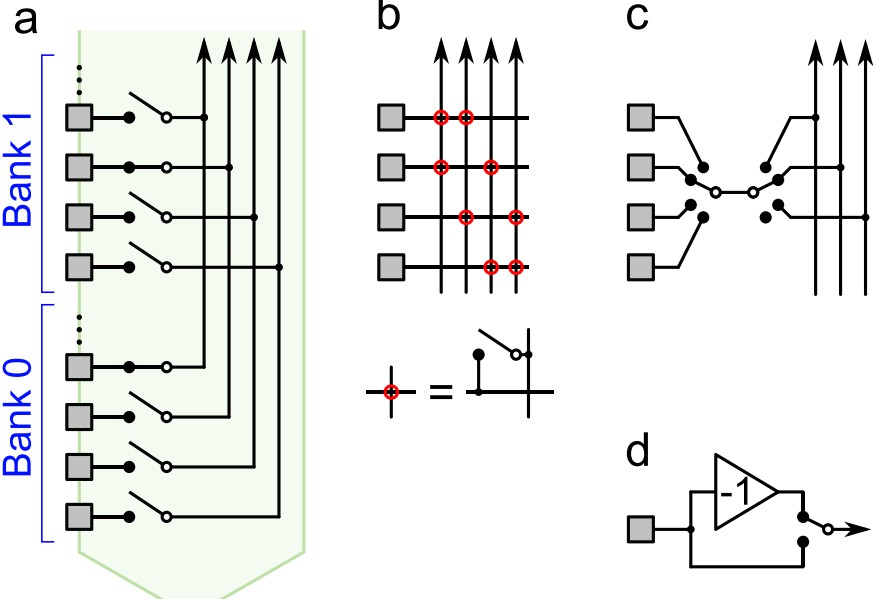

**Fig. 7 Hardware schemes for flexible connection between electrodes and wires. a** In the current Neuropixels array each electrode can be connected to just one wire with a controllable switch. **b** Two switches per electrode would allow a choice of 2 wires, enabling many more pooling configurations. **c** Because neighboring electrodes often carry redundant signals, one may want to choose just one from every group of 4. This switch circuit matches that choice with one of 3 (or no) wires. **d** An optional inverter for each electrode, controlled by a local switch.

half of the electrodes in a pool use the inverter, that helps to differentiate the spike shapes of different neurons. Because extracellular signals from cell bodies generally start with a negative voltage swing, this effectively doubles the space of waveforms that occur in the pooled signal. In turn, this would aid the spike-sorting analysis, ultimately allowing even more electrodes to share the same wire.

Of course each of these proposals comes with some cost, such as greater power use or added space required for digital logic. The overall design of a probe must take all these trade-offs into account. The several-fold gain in recording efficiency promised by electrode pooling should act as a driver in favor of fully programmable switches, but deciding on the optimal design will benefit from the close interaction between users and manufacturers.

*Software*. Electrode pooling will also benefit from further developments in spike-sorting algorithms. For example, a promising strategy is to acquire all the spike shapes present on the electrode array using split-mode recordings, compute the expected pooled-mode waveforms, and use those as templates in sorting the pooled signals. We have implemented this so-called "hot sorting" method in KiloSort2 and have shown that it can greatly increase the number of split-mode cells recovered in the pooled recordings (Fig. 5c). This idea may also be extended to cluster-based sorting algorithms, by guiding the initialization of the clustering step. Indeed, knowing ahead of time which waveforms to look for in the recording would help any spike-sorter. We expect this method will also improve the resolution of temporally overlapping spike waveforms.

As one envisions experiments with 10,000 or more recording sites, it becomes imperative to automate the optimal design of electrode pools, so that the user wastes no time before launching into pooled recording (Fig. 3). The pooling strategy can be adapted flexibly to the statistics of the available neural signals, even varying along the silicon shank if it passes through different brain areas. The user always has the option of recording select sites in conventional mode; for example, this might serve to sample local field potentials at a sparse set of locations. Designing an effective algorithm that recommends and implements the electrode switching based on user goals will be an interesting challenge.

**High-impact applications**. Finally, we believe that the flexible pooling strategy will be particularly attractive in chronic studies, where an electrode array remains implanted for months or years. In these situations, maintaining an updated library of signal waveforms is an intrinsic part of any recording strategy. Round-the-clock recording serves to populate and refine the library, enabling the design of precise spike templates, and effective separation of pooled signals. The library keeps updating in response to any slow changes in recording geometry that may take place.

A second important application for pooling arises in the context of sub-dural implants in humans. Here the sub-dural space forces a low-profile chip with minimal volume for electronic circuitry, whereas one can envision several slender penetrating electrode shafts with thousands of recording sites. We estimate that some devices that are now plausible (no published examples yet) will have an electrode-to-channel ratio near 25. Clearly one will want to record from more than 1/25 of the available sites, and electrode pooling achieves it without increased demand on electronic circuitry.

In summary, while the devices to maximize pooling benefits are not yet available, they soon may be. Consideration of pooling options would benefit the designers and users of these devices. The advantage of pooling grows naturally as the same tissue is recorded across sessions or time. The calculations and demonstrations reported here are intended to inspire professional simulations and the design of future devices for a variety of applications, including human implants.

## Methods
All analysis was performed with Matlab R2016b (Mathworks) and Python 3. All the quoted uncertainties are standard deviations.

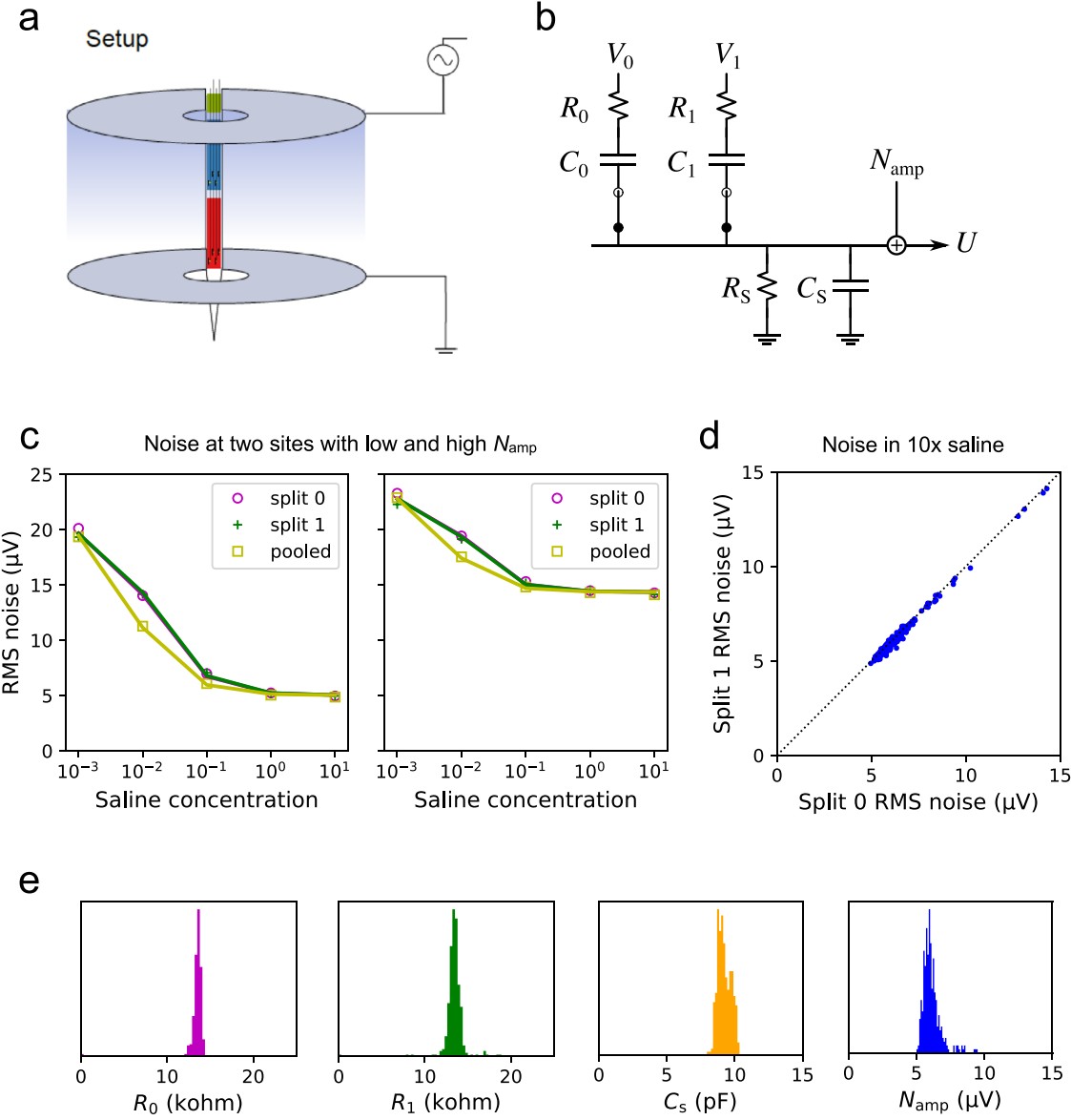

**Fig. 8 Methods for in vitro measurements of Neuropixels function. a** The probe is immersed in saline, with two annular electrodes producing an electric gradient along the shank. **b** Equivalent circuit model to understand signal and noise pooling for one wire of the array. **c** Measurements of noise only without an external field. RMS noise as a function of the saline concentration under three conditions of the switches: split recording from Bank 0, split recording from Bank 1, and pooled recording from both. Examples of two different wires, one with high, the other with low amplifier noise $N_{amp}$. **d** The noise at the highest saline concentration, recording from electrode 1 vs electrode 0. Each dot is for one of the 383 wires. This limiting noise is identical across the two electrodes on the same wire. **e** Histograms of the best-fit circuit parameters derived for each of the 383 wires on a pristine Neuropixels probe. $R_S$ is too large to be measured properly.

**Control of Neuropixels switching circuitry**. The Neuropixels 1.0 probe has 960 recording sites that can be connected to 384 wires via controllable switches. The conventional mode of operation (split mode) was to connect one electrode to one wire at a time. Electrode pooling was implemented by modifying the Neuropixels API and the GUI software SpikeGLX to allow connecting up to three electrodes to each readout wire.

**Neuropixels device measurements**. To characterize signal and noise pooling on the Neuropixels 1.0 array, we immersed the probe in a saline bath containing two annular electrodes to produce an electric field gradient (Fig. 8a). The electrolyte was phosphate-buffered saline (Sigma-Aldrich P4417; 1× PBS contains 0.01 M phosphate buffer, 0.0027 M potassium chloride and 0.137 M sodium chloride, pH 7.4, at 25 °C). We recorded from all 383 wires (recall that one wire is a reference electrode), first closing the switches in Bank 0 then in Bank 1, then in both banks (Fig. 3b).

One set of measurements simply recorded the noise with no external field applied. Then we varied the concentrations of PBS (by factors $10^{-3}$, $10^{-2}$, $10^{-1}$, 1, and 10), which modulated the conductance of the bath electrolyte in the same

proportions. For each of the 15 recording conditions (5 concentrations × 3 switch settings) we measured the root-mean-square noise on each of the 383 wires. Then we set to explain these 5 × 3 × 383 noise values based on the input circuitry of the Neuropixels device. After some trial-and-error we settled on the equivalent circuit in Fig. 8b. It embodies the following assumptions:

- Each electrode is a resistor $R_i$ in series with a capacitor $C_i$. The resistor is entirely the bath resistance, so it scales inversely with the saline concentration.
- The shunt impedance $Z_S$ across the amplifier input is a resistor $R_S$ in parallel with a capacitor $C_S$.
- The thermal noise from this R-C network and the voltage noise $N_{amp}$ from the amplifier and acquisition system sum in quadrature.

With these assumptions, one can compute the total noise spectrum under each condition. In brief, each resistor in Fig. 8b is modeled as a white-spectrum Johnson noise source in series with a noiseless resistor (Thevenin circuit). The various Johnson noise spectra are propagated through the impedance network to the output voltage $U$. That power spectrum is integrated over the AP band

(300–10,000 Hz) to obtain the total thermal noise. After adding the amplifier noise $N_{amp}$ in quadrature one obtains the RMS noise at the output $U$. This quantity is plotted in the fits of Fig. 8c.

The result is rather insensitive to the electrode capacitance $C_i$ because that impedance is much lower than the shunt impedance $Z_S$. By contrast, the bath resistance $(R_0, R_1)$ has a large effect because one can raise it arbitrarily by lowering the saline concentration. To set the capacitor values, we, therefore, used the information from the Neuropixels spec sheet that the total electrode impedance at 1 kHz is 150 kΩ,

$$C_i = \frac{1}{2\pi \cdot 1000 \ \text{Hz} \cdot \sqrt{(150\text{k}\Omega)^2 - R_i^2}} \quad (12)$$

We also found empirically that the shunt impedance is primarily capacitive: $R_S$ is too large to be measured properly and we set it to infinity. Thus the circuit model has only 4 scalar parameters: $R_0, R_1, C_S, N_{amp}$. Their values were optimized numerically to fit all 15 measurements. This process was repeated for each of the 383 wires. The fits are uniformly good; see Fig. 8c for examples.

As expected the thermal noise increases at low electrolyte concentration because the bath impedance increases (Fig. 8c). However, the noise eventually saturates far below the level expected for the lowest saline concentration. This reveals the presence of another impedance in the circuit that acts as a shunt across the amplifier input (Fig. 2a). We found that $Z_S \approx 20$ MΩ. Because the shunt impedance far exceeds the electrode impedances[2] (~150 kΩ), it has only a minor effect on signal pooling, which justifies the approximations made in Eq. (3).

The measured noise voltage also saturates at high saline concentration (Fig. 8c), and remains far above the level of Johnson noise expected from the bath impedance. That minimum noise level is virtually identical for the two electrodes that connect to the same wire, whether or not they are pooled, but it varies considerably across wires (Fig. 8d). We conclude that this is the amplifier noise $N_{amp}$ introduced by each wire's acquisition system (Fig. 2a).

Figure 8e shows the best-fit values of the 4 circuit parameters, histogrammed across all the wires on an unused probe. Note they fall in a fairly narrow distribution. The bath impedance of the electrodes (in normal saline) is ~13 kΩ, the shunt capacitance is ~10 pF, and the common noise $N_{amp}$ has a root-mean-square amplitude of ~6 μV integrated over the AP band (300–10,000 Hz).

These measurements were performed on both fresh and used Neuropixels devices, with similar results. On a device previously used in brain recordings the bath impedance of the electrodes was somewhat higher: 30 kΩ instead of 13 kΩ.

To measure the pooling coefficients we applied an oscillating electric field (1000 Hz) along the electrode array with a pair of annular electrodes (Fig. 8a). From the recorded waveform we estimated the signal amplitude by the Fourier coefficient at the stimulus frequency. Two different field gradients (called A and B) yielded two sets of measurements, each in the two split modes ($U_{0,A}$, $U_{1,A}$, $U_{0,B}$, $U_{1,B}$) and the pooled mode ($U_{P,A}$, $U_{P,B}$). For each of the 383 wires, we estimated the pooling coefficients of its two electrodes by solving

$$\begin{bmatrix} U_{0,A} & U_{1,A} \\ U_{0,B} & U_{1,B} \end{bmatrix} \begin{bmatrix} k_0 \\ k_1 \end{bmatrix} = \begin{bmatrix} U_{P,A} \\ U_{P,B} \end{bmatrix} \quad (13)$$

These mixing coefficients $k_0$ and $k_1$ express the recorded amplitude $U_P$ in terms of the recorded amplitudes $U_0$ and $U_1$,

$$U_P = k_0 U_0 + k_1 U_1 \quad (14)$$

whereas the pooling coefficients $c_0$ and $c_1$ (Eq. (2)) are defined relative to the input voltages $V_0$ and $V_1$, namely

$$U_P = c_0 V_0 + c_1 V_1 \quad (15)$$

The $U_i$ differ from the $V_i$ only by the ratio of electrode impedance to shunt impedance. Given the measured value of $Z_S \approx 20$ MΩ that ratio is <1%, a negligible discrepancy. So the measured $k_0$ and $k_1$ are excellent approximations to the pooling coefficients $c_0$ and $c_1$, which in turn reflect the ratio of the two electrode impedances (Eq. (2)).

**In vivo recording**. We used a Neuropixels 1.0 probe[2] to record neural signals from a head-fixed mouse (C57BL/6J, male, 9 months old). The probe entered the brain at 400 μm from the midline and 3.7 mm posterior from bregma at ~45° and was advanced for ~6 mm, which corresponded to all of Bank 0 and roughly half of Bank 1. This covered many brain areas, from the retrosplenial cortex at the top to the medial preoptic nucleus at the bottom. A detailed description of the mouse surgery, probe implantation, and post hoc histology and imaging of probe track can be found in a previous report[34]. All procedures were in accordance with institutional guidelines and approved by the Caltech IACUC, protocol 1656.

Once the probe was implanted, data were recorded in the following order: (1) split-mode in Bank 0 (i.e. all 384 wires connected to recording sites in Bank 0); (2) split-mode in Bank 1; (3) pooled-mode across Banks 0 and 1. Each recording lasted for ~10 min.

Following brain recordings, the array was cleaned according to recommended protocol by immersion in tergazyme solution and rinsing with water.

**Spike-sorting**. For "manual" spike-sorting of the in vivo recordings, we used KiloSort1 (downloaded from https://github.com/cortex-lab/KiloSort on Apr 10, 2018). We ran the automatic template-matching step; the detailed settings are available in the code accompanying this manuscript. This was followed by manual curation, merging units, and identifying those of high quality. These manual judgments were based on requiring a plausible spike waveform with a footprint over neighboring electrodes, a stable spike amplitude, and a clean refractory period. This was done separately for each of the three recordings (split-mode Bank 0, split-mode Bank 1, pooled-mode).

We implemented the "hot sorting" feature in KiloSort2 (downloaded from https://github.com/MouseLand/Kilosort2 on Mar 19, 2020). No manual curation was used in this mode, because (1) we wanted to generate a reproducible outcome, and (2) manual inspection is out of the question for the high-volume recordings where electrode pooling will be applied. We first sorted the two split-mode recordings and used their templates to initialize the fields W and U of rez2 before running the main template-matching function on the pooled recording (see the accompanying code for more details). Finally, the splits, merges, and amplitude cutoffs in Kilosort2 ensured that the final output contained as many well-isolated units as possible. We then selected cells designated as high quality (KSLabel of Good) by KiloSort2, indicating putative, well-isolated single neurons[35].

To elaborate on the internal operations of Kilosort2: Spike-sorted units were first checked for potential merges with all other units that had similar multi-channel waveforms (waveform correlation >0.5). If the cross-correlograms had a large dip (<0.5 of the stationary value of the cross-correlogram) in the range [-1 ms, +1 ms], then the units were merged. At the end of this process, units with at least 300 spikes were checked for refractory periods in their auto-correlograms, which is a measure of contamination with spikes from other neurons. The contamination index was defined as the fraction of refractory period violations relative to the stationary value of the auto-correlogram. The default threshold in Kilosort2 of 10 percent maximum contamination was used to determine good, well-isolated units.

Following spike sorting, we applied the matching algorithm based on cosine similarity (Fig. 5b) to determine how many cells identified in split recordings could be recovered from the pooled recording. This was compared with the results from "cold sorting", in which the pooled recording was sorted on its own, as well as to the conventional sorting that includes manual curation (Fig. 5c).

**Unmixing pooled signals**. After sorting the split and pooled recordings, we computed the average waveform of every cell. Specifically, for each cell we averaged over the first $n$ spikes, where $n$ was the lesser of 7500 or all the spikes the cell fired during the recording.

We then sought to identify every cell in the pooled recordings with a cell in the split recordings. This was done by the following procedure: Let $S$ denote a cell sorted from the split-mode recording ($S \in \mathcal{S}$) and $S_i$ its waveform at channel $i$. Although $i$ can range from 1 to 384 (the total number of wires available in the Neuropixels probe), we only focus on the 20 channels above and 20 channels below the channel with the largest amplitude ($i'$), i.e. $J = [i' - 20, i' + 20]$. We wish to find the cell $P$ from the pooled-mode recordings ($P \in \mathcal{P}$) that is closest to $S$. To do so, we compute the cosine similarity score for each pair $(S, P)$:

$$\Sigma(S, P) = \frac{\mathbf{S} \cdot \mathbf{P}}{||\mathbf{S}||||\mathbf{P}||} \quad (16)$$

where $\mathbf{S}$ and $\mathbf{P}$ are column vectors obtained by concatenating every $S_j$ and $P_j$ ($j \in J$), respectively, and $||\cdot||$ is the $\ell^2$ norm. $\Sigma$ is a $|\mathcal{S}|$-by-$|\mathcal{P}|$ matrix. We identify the largest element of $\Sigma$, which corresponds to the most similar pair of $S$ and $P$. We then update $\Sigma$ by removing the row and column of this largest element. This process gets iterated until every $P \in \mathcal{P}$ is given a best match. By manual inspection we found that pairs with similarity scores >0.9 were good matches.

**Estimating pooling coefficients in vivo**. Once each $P \in \mathcal{P}$ was assigned a match $S \in \mathcal{S}$, the pooling coefficient ($k$) was computed by solving the optimization problem below for each $i$ with a least squares method (mldivide in Matlab).

$$\text{eq : find}_c \ \arg\min_{k_i} ||P_i - k_i S_i|| \quad (17)$$

Sometimes a single recording site detected action potentials from multiple cells. As a result, its pooling coefficient could be estimated from the signal of each of these cells. Typically these estimates deviated from each other by <0.1. In these cases, we assigned the average of these values as the pooling coefficient of the recording site.

When two recording sites that share a wire in pooled mode each carry a significant signal, it enables the estimation of both of their pooling coefficients. Examples of such sites are shown in Figs. 5d–e (up to 50 pairs in Banks 0 and 1).

**Simulation**

*Generating simulated data*. We simulated extracellular voltage signals on 12 groups of 4 local electrodes ("tetrodes"). Each time series was sampled at 30,000 samples/s and extended over 600 s. After combining signal and noise as described below, the time series were filtered with a passband of 300–5000 Hz.

Each tetrode carried spikes from a single unit. The spike waveform of the unit was chosen from an actual mouse brain Neuropixel recording, with a different

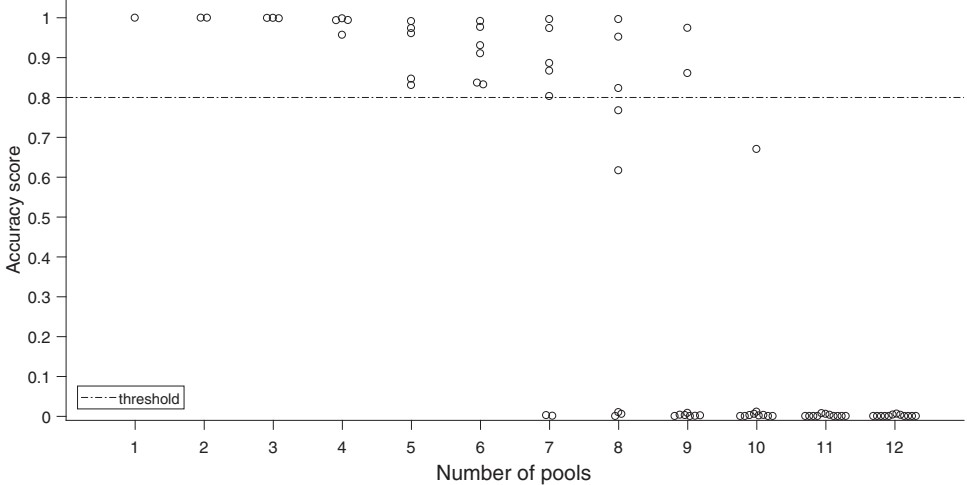

**Fig. 9 Accuracy scores of one "standard" condition simulation, Related to Fig. 6.** Units with accuracy score> 0.8 were counted as recovered.

waveform on each tetrode. Within a tetrode, one electrode chosen at random carried this spike at the nominal peak-to-peak amplitude, $V$ (Fig. 6b). On the other three electrodes, the spike was scaled down by random factors drawn from a uniform distribution over [0,1]. The spike train was simulated as a Poisson process with a forced 2-ms refractory period, having an average firing rate $r$ (Fig. 6b).

Three sources of noise—biological noise $N_{bio}$, thermal electrode noise $N_{the}$, and common amplifier noise $N_{com}$—were generated as gaussian processes. The quoted noise values (Fig. 6b) refer to root-mean-square amplitude over the 300–5000 Hz passband. Thermal noise was sampled independently for each electrode, but the biological noise was identical for electrodes within a tetrode, given that they likely observe the same background activity.

Electrode pooling across $M$ tetrodes was implemented by combining the voltage signals of the corresponding electrode on each tetrode, resulting in signals on four wires. In the process each electrode signal was weighted by $1/M$, then the amplifier noise was added to the resulting average. Amplifier noise was sampled separately for each wire.

Tetrodes were added to the pool in a sequence determined by the spike shape of their units. We started with the two most dissimilar units as determined by the cosine similarity of their spike waveforms. Then we progressively added the unit that had the lowest similarity with those already in the pool.

*Sorting simulated data*. The simulated 4-wire time series were sorted using Kilo-Sort2; detailed configuration settings are available in the code accompanying this paper. We found it necessary to turn off the "median voltage subtraction" during preprocessing, because that feature somehow introduced artifacts in the 4 voltage traces. This did not occur when processing electrode array data with many channels, for which the algorithm is intended. We note that an effective means of subtracting the common signal across wires may help suppress the biological noise and lead to better sorting results.

When large numbers of tetrodes were pooled the signal-to-noise ratio dropped to the point where KiloSort2 could not form templates in the preprocessing step. Under those conditions, we report zero units recovered (Fig. 6b).

*Scoring simulated data*. Following previous reports[26,36], the spike times of the sorted units and the ground truth units were matched and compared using the confusion matrix algorithm from ref. 36. We set the acceptable time error between sorted spikes and ground-truth spikes at 0.1 ms. Then we counted the number of spike pairs with matching spike times, $n_{match}$, the number of unmatched spikes in the ground-truth unit, $n_{miss}$, and the number of unmatched false-positive spikes in the sorted unit, $n_{fp}$.

To assess the quality of the match between ground-truth and sorted units we adopted the *Accuracy* definition in ref. 26:

$$\text{Accuracy} = \frac{n_{match}}{n_{match} + n_{miss} + n_{fp}} \quad (18)$$

Figure 9 shows the accuracy distribution obtained for various degrees of pooling. Sorted units with accuracy >0.8 were counted as "recovered" from the pooled signal. For each parameter set we ran the simulation three times, randomizing the noise and the spike times. Results from the three runs are reported by mean ± SD (Fig. 6b).

**Reporting summary**. Further information on research design is available in the Nature Research Reporting Summary linked to this article.

## Data availability
All data relevant to the reported results are available in a public repository: https://github.com/markusmeister/Electrode-Pooling-Data-and-Code. An archived version is available from CaltechDATA: https://doi.org/10.22002/D1.2032.

## Code availability
All code used to obtain the reported results are available in a public repository: https://github.com/markusmeister/Electrode-Pooling-Data-and-Code. An archived version is available from CaltechDATA: https://doi.org/10.22002/D1.2032.

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

## Acknowledgements
This work was supported by a grant to M.M. from NINDS (5R01NS111477) and an award to M.M. from the Tianqiao and Chrissy Chen Institute for Neuroscience. Y.L.N. was supported by the Taipei Veterans General Hospital – National Yang-Ming University Physician Scientists Cultivation Program, No.103-Y-A-003.

## Author contributions
K.H.L., Y.L.N., and M.M. conceived of the study. K.H.L. and Y.L.N. did experiments and simulations. K.H.L., Y.L.N., and M.M. analyzed the resulting data. J.C., B.K., J.P., M.P., and T.D.H. wrote software for acquisition and analysis, and advised on the use of Neuropixels. K.H.L., Y.L.N., and M.M. drafted the article. All authors edited the manuscript.

## Competing interests
The authors declare no competing interests.
