## [Peer Review File · Nature Communications]

Electrode pooling can boost the yield of extracellular recordings with switchable silicon probesReviewers' comments:

Reviewer #1 (Remarks to the Author):

This is a very well written paper with a considerable amount of effort put into theoretical estimation, experimental verification and simulation. This is certainly a carefully written draft. However, my main concern is the applicability of this method in real-life. To me, it looks like an excellent theoretical exercise but with a big question mark on who will want to use such a method. The main idea of pooling, as authors have noted, is similar to compression, in fact, close to compressed-sensing. It is widely known that even though neural data compression is a rather mature field now, very few neuroscientists actually want their data to be compressed right at the hardware. There are various reasons why neuroscientists do not want to 'believe' in a pre-compressed data. The authors here have provided no evidence to show that this attitude is changing. In fact, they have provided no reference for any paper on hard-wired based data compression at all.

The other important fact is that need for a split mode recording and various other calibration (private-noise, pooling coefficient etc.) necessary before the start of the actual recording. There seems to be a very extended calibration regime much beyond what can be done in a commercial device. The user themselves will have to do this since they need to be done for an individual probe. The calibration is also likely to shift over time.

There is also a 'learning' phase where the user needs to create a database of split-mode recording. This seems another issue that has already been a bottleneck for hardware data compression. If such databases can be relied upon for a long recording, then data compression would be rather easy. What about probe movement, glial formation etc. If the learning needs to be done repeatedly this also comes with a question of how often and how does one know when?

The algorithm for optimal pooling is a good idea and provides a lot with a flexibility to the user. However, this creates a dilemma for the hardware designer. What is the best design that covers the maximum user base?

It seems for the best pooling arrangement a rather precise measurement of the private noise and the common noise is necessary. Even they are calibrated once, noise measurement in such high-density analog circuits is certainly going to have a lot of uncertainty. One can only say that the noise has an upper limit, at best.

Explanation of the 3rd bank (bank 2) seems unnecessary and confusing (line 225-233). The whole point is to explain the pooling of 0+1, bringing in bank2 (which is hardly used) is not very useful. The section regarding future hardware implementation with more switches is not clear to me. It seems at least 2 or 3bit register is necessary for providing a wide variety of pooling option. This itself, along with the switches and the local interconnects can significantly increase the size of the electronics per site. Furthermore, it could also reduce the number of available wires for global connection. Without an understanding of the layout, the claims of reduced probe width look quite vague.

The section starting in line 421 seem to mention a digital inverter where an analog inverter is necessary. This could further increase the area and add a new variable on mismatch among sites. The claims of the paper on devising a new method of high-density recording from neural probes is well supported and is indeed an interesting addition to the field.

The paper starts with a plea to hardware designers, which by itself makes sense. But the hardware designer is now left with a much wide range of variables to play with. It is an understatement to say that there is hardly any consensus between neurophysiologist on the dimensions of the existing probes (trade-off between electrode density, channel count, width, length ...). Starting with a completely new set of choices (for pooling) will make the field of hardware even more open. On top of this, the acceptability of such 'compressed' data by the wider neuroscicne community is a question by itself.

Reviewer #2 (Remarks to the Author):

Please find my comments on the submission entitled: Electrode pooling: boosting the yield of extracellular recordings with switchable silicon probes.

I have some basic concerns regarding the “quick analysis to extract the relevant data that will inform the pooling process.”

If not automatized, in my opinion, this step may take long time (even hours), especially if the electrode/wire ratio is high, where the electrode pooling method should have a bigger advantage. The short period of acquisition in „split mode“ must be at least several minutes long per electrode group to capture enough spikes of low firing single units and to make a reliable spike sorting possible.

In the case of the probe described in Section 5.2.1, preliminary data collection from 11 electrode groups would take in my estimation, nearly 3 hours. After that, the spike sorting and a minimal manual curation would add another 1-2 hours to the preprocessing step. Although it is not obvious how much time would it take to decide which electrodes to pool based on the results of the preprocessing, but I think the duration of this period would also be significant: the pooling decision and the electrode selection step might take about an hour or more (depending on the number of electrodes) if not assisted by automatic or semiautomatic processes, especially if we also take into account the four principles described in the manuscript, since these need additional processing steps.

By adding the above time figures, it seems, that the initial acquisition, preprocessing and electrode selection steps might take hours which makes the practical use of electrode pooling method difficult, especially in the case of shorter (e.g. acute) experiments.

In addition, the electrode drift might further complicate this procedure: in the case of faster drifts when significant changes in spike waveforms and amplitudes happens over minutes, the use of electrode pooling might be not possible, or only with a low performance, while slower drifts (changes over hours) the entire acquisition-preprocessing-electrode selection step will have to be repeated every 2-5 hours.

Although the authors mention that these steps need to be automated to decrease their time cost, they do not detail how to speed up these steps.

I think that a more detailed spike sorting section is needed, here are my questions:

What were the criteria to select high quality units in the case of sorting with Kilosort1?

In the case of „hot sorting“ and „cold sorting“ units „only designated as high quality (KSLabel of Good) by KiloSort2“ were selected. Why there was no manual inspection here?

Authors checked the matched units, but what about the remaining units sorted in pooled mode?

Might these be new units not sorted in split recordings (e.g. new units appearing due to electrode drift, or low firing units not firing during the initial split recording) or units which could not be matched unequivocally between split and pooled recordings? I think the spike sorting results should be analysed in more detail.

What were the unit yields for each split recording separately?

Also, it would be interesting to provide quality metrics for the single unit clusters and compare the quality of single units between split and pooled recordings (e.g

https://allensdk.readthedocs.io/en/latest/_static/examples/nb/ecephys_quality_metrics.html).

Other possible difficulties are listed below:

In my opinion the method may suffer during synchronized brain activity (e.g. epilepsy), when overlapping spikes are abundant.

The proposed procedure will lose a significant fraction of lower amplitude units which can be clustered in split recordings.

Higher real amplitude units will be recorded with a lower amplitude, which in turn might decrease the accuracy/reliability of the sorting and the quality of clusters.

Some spikes in bursts (mostly the smaller last ones) might be missed in pooled recordings because of the amplitude decrease during electrode pooling

If we pool multiunit activity or local field potentials recorded by distant electrodes, these signals cannot be unmixed later, so the procedure is not applicable for these recording modalities.

I think the authors should analyse more recordings (at least 3 recordings from different brain

areas) to make the results more robust.

I am aware that Neuropixels probe recordings might not be the best to show the advantages of electrode pooling (i.e. higher unit yield/wire), nonetheless the increase in the unit yield seems not so high (e.g. in the case of the „hot sort“: 332 vs. 372 units for split and pooled recordings, respectively). I would suggest emphasizing in the manuscript that we can record from these units simultaneously in the case of pooled recordings, while in the case of split recordings we have to obtain two separate recordings to get this unit yield.

The authors should provide more detail on the probe cleaning procedure.

Reviewer #3 (Remarks to the Author):

The authors present a novel method to boost the yield of extracellular recordings called electrode pooling. This involves connecting the same wire to multiple switchable recording sites instead of just one electrode at a time, which is the normal practice. The authors neatly synthesize results from an analytical model, experimental data with a modified Neuropixel, and simulations to explore the merits of electrode pooling. The paper was well written and easy to understand for the most part. The main strengths of the study are in terms of technical innovation and potential significance for switchable silicon probe makers such as the Neuropixel consortium. Despite these strengths the study was perceived to contain significant limitations which lowered overall enthusiasm. As detailed below, the main concern was lack of compelling evidence that pooling is either beneficial or practical to implement. The reviewer found a number of areas which could be improved with revisions, although it is unclear if the major concerns noted below are surmountable. The reviewer feels this study is still important but perhaps not in the way the authors intended. In the reviewer's opinion it shows that electrode pooling is unlikely to be a viable general-purpose method...but, that it may be useful in certain specialized situations involving sparse, high spike amplitude and low noise activity, and when perfect unmixing of data is not required. It is useful for the Neuropixel community to know these limitations in order to avoid false starts. Additionally, there may be some niche applications where the benefits outweigh the tradeoffs.

Major concerns:

1. The biggest concern was that the paper does not provide compelling evidence that electrode pooling offers clear benefits in terms of extracellular recording yield. The authors are commended for exploring pooling using diverse approaches including a theoretical model, experiment with a modified Neuropixel, and simulated data. Unfortunately, none of those results were convincing that electrode pooling provides significant benefits in typical extracellular recording conditions. First, pooling only seems effective for high amplitude spikes (for example in line 356 they mention that 380 microvolts was in the 90th percentile of spike amplitudes from the Allen Institute database). Yet the same database shows that most spikes have a much lower amplitude closer to 100 microvolts. Second, under the most ideal conditions the neuron/wire ratio was 8 (Figure 2), which would be a respectable improvement. However, this ratio drops close to ~ 1 when the conditions become less favorable and more in line with typical spike and noise amplitudes. The authors speculated in line 365 that if common noise could be reduced by a factor of 2 then the optimal pool size would increase. But throughout the paper they assumed that the only source of common noise comes from the amplifier (N_{amp}), and that biological noise N_{bio} is unique to each electrode. This seems to be an optimistic assumption, as at least part of N_{bio} may be common to many electrodes. Thus, the authors' pooled noise estimates may have been consistently underestimated, and it is unclear how a factor of 2 noise reduction would ever materialize. Third, it seems that efforts to unmix pooled signals in Figure 5 was only moderately effective. Overall, it appears pooling may provide some modest advantages for only a select group of cells with high spike amplitude and low noise. But for the large majority of spikes measured in a recording, according to the results presented there is no unambiguous benefit to pooling, and it may even be detrimental for obtaining single cell spiking information.

2. A related concern was that the proposed workflow for pooling described in Figure 3 seems quite complex, raising substantial concerns about the practicality of using this approach. Additionally,

the four principles for choosing which electrodes to pool (lines 192-196) seem quite restrictive, further raising concerns that a large percentage of electrodes will be ineligible for pooling. To the authors' credit, they wrote a thoughtful Discussion section that outlined some of the possible opportunities for using electrode pooling in future generations of Neuropixel probes. They mentioned that pooling is likely to become more relevant as the number of recording sites greatly exceeds than the number of available wires. For this to be true, many of the problems or limitations noted in the first comment would need to be addressed, but the path to achieving that is not clear. Thus it seems uncertain if the benefit/tradeoff ratio is great enough to ever justify using this as a standard approach.

Minor comments:

3. Line 80: "...are summed..." Suggest changing to "averaged" as this is an important distinction as shown by equation 3.

4. Lines 94-95: "Sometimes a second neuron.....". Seems more appropriate to make this a general statement, eg "Sometimes multiple neurons (as many as x based on literature)....."

5. Figure 2: The red dots based on the experimentally derived values of alpha and beta are useful. It is suggested that authors change the dots into a shaded zone based on a range of experimentally derived values to give a better sense of what conditions are likely to be encountered. The red dot in panel 2C is based on highly optimistic alpha and beta values, and it would be useful to show a bigger range of values.

6. Equation 4: Recommend renaming private noise as independent or uncorrelated noise, and renaming the thermal noise as N_{th} rather than N_{ele} to avoid confusion with the electronic noise.

7. Line 135: The authors assume that N_{bio} is independent ("private") but it seems more conservative to assume it is common or at least partially common. By assuming N_{bio} is entirely private their beta values in equations 9 & 10 may be optimistically high, inflating the optimal M_{max} shown in Figure 2C. As noted in the next comment, since the authors have access to experimental data with pools of 2 and 3 electrodes, it should be possible to compare noise levels in split and pooled mode data.

8. The noise characterization in Figures 4 and 5E could be improved. First, it would be useful to put all noise-related graphs in a single figure rather than split them between two figures. Second, the authors are asked to show 3 histograms corresponding to the rms value of the 3 main noise types (N_{bio} , N_{amp} , $N_{thermal}$). Third, an attempt should be made to compare how the total noise measured in vivo (N_{tot}) looks in split mode versus pooled mode using data from the modified Neuropixel probe. Ideally this might allow the authors to use Equation 6 or 7 to estimate the total amount of N_{pri} and N_{com} . In turn, this might provide insight into whether N_{bio} is predominantly "private" or "common".

9. Lines 188-190: "Finally, for every electrode one gets the private noise N_{pri} . The common noise N_{com} can be assessed ahead of time, because it is a property of the recording system." As noted above this assumes that N_{bio} is part of the private noise. It seems more rigorous to rephrase that sentence to say that for every electrode one gets the total noise. The amplifier and thermal noise can be measured ahead of time because they are basic properties of the recording system. With this information, one can calculate N_{bio} for every electrode.

10. In lines 232-233, do the authors mean that the pooling function allows pooling of 2 or 3 wires, with at most 1 wire per bank? This was a bit unclear. The authors also mention that the pooling coefficient was measured at ~ 0.5 . This suggests they only used pools of 2 but never 3 wires, is that correct? If so please clarify in the text. Also, please clarify if the selection of 2 wires (one per bank) is fixed (if electrode A from bank 0 can only be pooled with electrode A from bank 1) or customizable (if electrode A from bank 0 can be pooled with any of the 384 electrodes from bank 1). Judging by some of the data and from line 304 it seems likely that the selection is fixed but it would be helpful to provide these details in Section 3.1 of the paper.

11. Figure 5: the presentation of some results in this figure was confusing and appeared to lack some crucial details. Some important information about the number of cells was delegated to Table 1 in the supplementary data, but this is crucial information that belongs in the main text and preferably within Figure 5. Additionally, panel 5B was difficult to interpret for several reasons. First, were the results obtained with cold sorting? If so the authors are requested saying so in the caption. Second, it is suggested changing the color scheme to increase contrast around cosine of 0.9. With the current color scheme it is difficult to distinguish the various red shades. Third, it is suggested somehow trying to label the 122 matched cells or grouping them together in the matrix. In panel 5C it is suggested renaming the y axis label to "Matched cells" and changing the plot style to percentage (ratio of matched cells to total original split cells) rather than absolute number. In terms of additional details it would be valuable to show a histogram of the distribution of spike amplitudes in split mode, which is useful for estimating S_{max}/S_{min} .

12. Figure 6: For the authors to make a persuasive argument that pooling is beneficial, they would need to demonstrate that pooling provides robust gains under a wide range of possible conditions. Unfortunately, in this section the exploration of various parameters that influence performance was very limited. In particular, they only explored two spike amplitudes, with the lower bound being 205 microvolts. However, 205 microvolts is still on the high end of the amplitude range, with the mean and median amplitude being closer to 100 microvolts (see Siegle biorxiv 2019, Supplementary Fig. 4). How would the simulated pooling performance look with these more typical spike amplitudes? Second, the authors use the Accuracy metric defined by Magland 2020, but do not report the accuracy values. It would be useful to include this information because their criterion for matched cells (accuracy greater than 0.8) seems a bit arbitrary.

Reply to reviewers:

We thank the reviewers for thoughtful comments on this article. The full reviews are reproduced below with some section numbers inserted for convenience. The reviewers seem to appreciate the work as an interesting contribution to the field of multi-electrode array recording [1.1, 1.8, 3.0]. They voice some skepticism whether this method will offer substantial benefits [2.3, 2.4, 2.5, 3.1]; whether it will be too complex to implement [1.3, 2.1, 3.2]; whether the user community will want it [1.2]; and whether hardware designers will incorporate it [1.4, 1.9]. Below we respond to these main concerns with a summary reply, and then to the detailed comments.

Will the users accept a device with hardware compression and will hardware designers build the concept into new devices? [1.2, 1.4, 1.9]

These two issues are obviously linked: Manufacturers will not invest substantial sums in a new hardware design unless there is a clear demand. But users cannot even express such a demand as long as they are unaware of the possibilities. Our paper is an attempt to break through this chicken-and-egg dilemma, using calculations and experiments with a partially useful device. Publication of this proposal would be a way to get the ball rolling. We added text to the introduction to clarify the paper's purpose (Line 86).

Regarding potential users, Reviewer 1 questions whether neuroscientists 'believe' in hardware compression. The method we propose requires no belief, because the user retains full control over the degree of compression. One option is always to use the conventional switch setting with one electrode per wire, no compression, and record from parts of the array in the old-fashioned mode. We added text to make this clear (Line 443).

However, we predict that users will find many situations where they can easily record from more neurons by pooling electrodes. They may do so flexibly on some electrodes, and refrain from pooling on others. As a concrete example, a newly disclosed device, Neuropixels 2.0, has 5120 electrodes but only 384 channels [Steinmetz et al 2021]. In conventional mode, the user will be limited to recording from 1/13 of the device. There will be many cases where spikes with high signal-to-noise ratio are available on more electrodes than there are channels. Electrode pooling can resolve this problem. Indeed, Steinmetz et al 2021 already explored a version of pooling across banks of electrodes, while citing our preprint of the current manuscript. The revised manuscript includes these recent developments (Line 398).

Regarding hardware designers, Reviewer 1 asks "What is the best design that covers the maximum user base?" This is exactly the discussion we would like to stimulate. It can only take place once the user community sees the potential opportunity. Users of conventional arrays may want to simulate how well they could do under various pooling schemes using their own database of recordings. Our published code should help in exploring that.

Is the method too complex for the user to implement? [1.3, 2.1, 3.2] As our

paper describes, the method requires making informed choices about which electrodes to combine in a pool, and these depend on the real-time recording conditions. Obviously that task will be performed by an automated algorithm that samples sequentially from every electrode in the array, performs a quick analysis of the signals, and then constructs effective electrode pools. The time required for this is a small fraction of the recording time.

Reviewer 2's estimates here are too pessimistic. Taking the newly announced Neuropixels 2.0 device with 5120 electrodes and 384 channels: A sample of 5 minutes each from 13 groups of electrodes takes a bit over an hour. Spike sorting and associated analysis can proceed in parallel with that and requires no additional time. Obviously all these steps are automated; for example, we used fully automated spike-sorting in our analysis, and these methods are improving steadily.

Note that these same steps also apply in conventional recording mode. The user still has to choose 384 electrodes out of the 5120 options, and will want to scan the whole array to see where the good signals are. That same scan can serve to decide on a more sophisticated switch setting for electrode pooling. Some of the concerns listed by the reviewers (electrode drift, epileptic activity) apply equally to conventional mode recording, and their respective solutions will be beneficial in pooled mode as well.

The users of high-count electrode arrays are already familiar with the need for automation. For better or worse, the days of watching a single trace on the oscilloscope in real time are gone (the senior author still does it at times for nostalgic reasons or for debugging). Instead, both recording and post-processing are managed via software that controls the acquisition electronics. The algorithms we advocate to steer electrode pooling will simply become part of that software suite.

We added some text to emphasize the importance of automation and an estimate of the time required for calibration (Line 221, Line 441).

Will the method offer substantial benefits? [2.3, 2.4, 2.6, 3.1b, 3.1c] As Reviewer 2 states, some low-amplitude spikes will be missed because of pooling [2.4]. On the other hand, without electrode pooling many high-amplitude spikes will be missed simply because there is no recording channel available. For example in Neuropixels 2.0 only one of 13 electrodes can be used in the conventional mode. The revised experimental section has a new figure showing the probability that a unit is recovered from the pooled signal as a function of its spike amplitude (Fig 5C). Even under the constrained conditions for electrode pooling we found that pooling is beneficial for spike amplitudes 100 μV and higher. This shows that the approach is not limited only to the largest spikes. Reviewers 2 and 3 question our assumptions in this analysis [2.3, 3.1b, 3.7], especially whether biological noise is uncorrelated across electrodes. As we state in the manuscript, this consideration recommends pooling of electrodes located some distance apart. Neuropixels 1.0 and 2.0 span distances up to 10 mm, traversing multiple areas of the rodent brain: There should be adequate room to find electrode sites a few mm apart with independent noise. In the revised

version we added a figure that confirms this expectation in our prototype recordings (Fig 4F).

Finally, we failed to advertise properly our results of separating pooled signals (Fig 5) [3.1c]: As Reviewer 2 points out [2.6] we were already able to record more units in pooled mode than during the average split mode recording. We have simplified and revised this section to show that electrode pooling works not just for the largest action potentials, and that it can boost the yield even for small spikes (Line 284).

Full text of reviews with responses not covered above

Reviewer #1 (Remarks to the Author):

1.1. This is a very well written paper with a considerable amount of effort put into theoretical estimation, experimental verification and simulation. This is certainly a carefully written draft.

However, my main concern is the applicability of this method in real-life. To me, it looks like an excellent theoretical exercise but with a big question mark on who will want to use such a method.

See summary reply above.

1.2. The main idea of pooling, as authors have noted, is similar to compression, in fact, close to compressed-sensing. It is widely known that even though neural data compression is a rather mature field now, very few neuroscientists actually want their data to be compressed right at the hardware. There are various reasons why neuroscientists do not want to 'believe' in a pre-compressed data. The authors here have provided no evidence to show that this attitude is changing. In fact, they have provided no reference for any paper on hard-wired based data compression at all.

That was because we do not propose hard-wired data compression. The entire scheme is based on software-controlled switches, which include the option of "no compression". We have added some text to emphasize this point including references to examples of hardware compression (Line 388).

1.3. The other important fact is that need for a split mode recording and various other calibration (private-noise, pooling coefficient etc.) necessary before the start of the actual recording. There seems to be a very extended calibration regime much beyond what can be done in a commercial device. The user themselves will have to do this since they need to be done for an individual probe. The calibration is also likely to shift over time.

There is also a 'learning' phase where the user needs to create a database of split-mode recording. This seems another issue that has already been a bottleneck for hardware data compression. If such databases can be relied upon for a long recording, then data compression would be rather easy. What about

probe movement, glial formation etc. If the learning needs to be done repeatedly this also comes with a question of how often and how does one know when?

The revised text gives a more detailed account of what the early learning phase would entail (Line 221). We also highlight in the discussion the application of the method during chronic recordings, where one can continuously update the library of waveforms available on every electrode and adjust the pooling strategy accordingly (Line 450).

1.4. The algorithm for optimal pooling is a good idea and provides a lot with a flexibility to the user. However, this creates a dilemma for the hardware designer. What is the best design that covers the maximum user base? It seems for the best pooling arrangement a rather precise measurement of the private noise and the common noise is necessary. Even they are calibrated once, noise measurement in such high-density analog circuits is certainly going to have a lot of uncertainty. One can only say that the noise has an upper limit, at best.

See summary reply above.

1.5. Explanation of the 3rd bank (bank 2) seems unnecessary and confusing (line 225-233). The whole point is to explain the pooling of 0+1, bringing in bank2 (which is hardly used) is not very useful.

Revised as suggested (Line 236).

1.6. The section regarding future hardware implementation with more switches is not clear to me. It seems at least 2 or 3bit register is necessary for providing a wide variety of pooling option. This itself, along with the switches and the local interconnects can significantly increase the size of the electronics per site. Furthermore, it could also reduce the number of available wires for global connection. Without an understanding of the layout, the claims of reduced probe width look quite vague.

The discussion mentions these tradeoffs. Properly evaluating them will require interaction between silicon engineers and prospective users. Again, we hope to initiate debate of these issues. We added some text to the discussion along those lines (Line 426).

1.7. The section starting in line 421 seem to mention a digital inverter where an analog inverter is necessary. This could further increase the area and add a new variable on mismatch among sites.

The paper we cited describes analog CMOS inverters. This is now clarified in the text (Line 419).

1.8. The claims of the paper on devising a new method of high-density recording

from neural probes is well supported and is indeed an interesting addition to the field.

The paper starts with a plea to hardware designers, which by itself makes sense.

See summary reply above.

1.9. But the hardware designer is now left with a much wide range of variables to play with. It is an understatement to say that there is hardly any consensus between neurophysiologist on the dimensions of the existing probes (trade-off between electrode density, channel count, width, length ...). Starting with a completely new set of choices (for pooling) will make the field of hardware even more open. On top of this, the acceptability of such 'compressed' data by the wider neuroscience community is a question by itself.

See summary reply above.

Reviewer #2 (Remarks to the Author):

Please find my comments on the submission entitled: Electrode pooling: boosting the yield of extracellular recordings with switchable silicon probes.

2.1. I have some basic concerns regarding the “quick analysis to extract the relevant data that will inform the pooling process.”

If not automatized, in my opinion, this step may take long time (even hours), especially if the electrode/wire ratio is high, where the electrode pooling method should have a bigger advantage. The short period of acquisition in „split mode” must be at least several minutes long per electrode group to capture enough spikes of low firing single units and to make a reliable spike sorting possible.

In the case of the probe described in Section 5.2.1, preliminary data collection from 11 electrode groups would take in my estimation, nearly 3 hours. After that, the spike sorting and a minimal manual curation would add another 1-2 hours to the preprocessing step. Although it is not obvious how much time would it take to decide which electrodes to pool based on the results of the preprocessing, but I think the duration of this period would also be significant: the pooling decision and the electrode selection step might take about an hour or more (depending on the number of electordes) if not assisted by automatic or semiautomatic processes, especially if we also take into account the four principles described in the manuscript, since these need additional processing steps.

By adding the above time figures, it seems, that the initial acquisition, preprocessing and electrode selection steps might take hours which makes the practical use of electrode pooling method difficult, especially in the case of shorter (e.g. acute) experiments.

In addition, the electrode drift might further complicate this procedure: in the case of faster drifts when significant changes in spike waveforms and amplitudes happens over minutes, the use of electrode pooling might be not possible, or only with a low performance, while slower drifts (changes over hours) the entire acquisition-preprocessing-electrode selection step will have to be repeated every 2-5 hours.

Although the authors mention that these steps need to be automated to decrease their time cost, they do not detail how to speed up these steps.

See summary reply above. The revised text addresses this issue explicitly (Line 221).

2.2.

I think that a more detailed spike sorting section is needed

We provide these details in the revised methods section (Line 553).

here are my questions:

What were the criteria to select high quality units in the case of sorting with Kilosort1?

We chose high-quality units based on manual inspection of the spike waveform (typical extracellular action potential? footprint on neighboring electrodes?), a stable spike amplitude, and a clean refractory period. These are common criteria for manual inspection.

In the case of „hot sorting” and „cold sorting” units „only designated as high quality (KSLabel of Good) by KiloSort2” were selected. Why there was no manual inspection here?

No manual curation was used in this mode, because (1) we wanted to generate a reproducible outcome, and (2) manual inspection is out of the question for the high-volume recordings where electrode pooling will be applied. The revised methods section gives more detail on Kilosort2’s internal operations (Line 569).

Authors checked the matched units, but what about the remaining units sorted in pooled mode? Might these be new units not sorted in split recordings (e.g. new units appearing due to electrode drift, or low firing units not firing during the initial split recording) or units which could not be matched unequivocally between split and pooled recordings?

Many of the pooled cells that were not matched indeed had low firing rates and may have been missed in the split recording. We have now revised this analysis to be more conservative: we only report those units from the pooled recording that had a match in the split recordings (Line 284, Fig 5C).

I think the spike sorting results should be analysed in more detail.
What were the unit yields for each split recording separately?
Also, it would be interesting to provide quality metrics for the single unit clusters and compare the quality of single units between split and pooled recordings (e.g. https://allensdk.readthedocs.io/en/latest/_static/examples/nb/ecephys_quality_metrics.html).

Other possible difficulties are listed below:

We feel that at this stage of the game such detailed assessments will not be helpful to the reader. Our goal is to demonstrate that spike trains can be unmixed in principle, even with very constrained pooling choices. Some day, when a device with flexible switches is available, it will make sense to evaluate performance in great detail and see how it varies with pooling strategy.

2.3. In my opinion the method may suffer during synchronized brain activity (e.g. epilepsy), when overlapping spikes are abundant.

The revised text mentions this early on (Line 100). Those events are a challenge for conventional recording as well. If such activity is the target of the study one would choose the pooling configuration accordingly. Again, you can always turn pooling off.

2.4. The proposed procedure will lose a significant fraction of lower amplitude units which can be clustered in split recordings.
Higher real amplitude units will be recorded with a lower amplitude, which in turn might decrease the accuracy/reliability of the sorting and the quality of clusters. Some spikes in bursts (mostly the smaller last ones) might be missed in pooled recordings because of the amplitude decrease during electrode pooling.

Again, this is also an issue in conventional mode recording if spikes drop below the recording threshold. But on many occasions one has excess signal-to-noise that can be used for signal pooling. See summary reply above.

2.5. If we pool multiunit activity or local field potentials recorded by distant electrodes, these signals cannot be unmixed later, so the procedure is not applicable for these recording modalities.

Correct. LFPs can be sampled at lower spatial resolution, so one can designate some small fraction of electrodes for single-site recording of LFPs. We added this to the revised text (Line 445).

2.6. I think the authors should analyse more recordings (at least 3 recordings from different brain areas) to make the results more robust.

We are currently unable to perform further experiments, owing to constraints on laboratory use. However, we also do not believe that this is a bottleneck here. As the reviewer states, Neuropixels 1.0 is not the device one will ultimately use for electrode pooling. So the objective here is simply to demonstrate that spikes from different electrodes can be pulled apart not only in theory but also in practice.

I am aware that Neuropixels probe recordings might not be the best to show the advantages of electrode pooling (i.e. higher unit yield/wire), nonetheless the increase in the unit yield seems not so high (e.g. in the case of the „hot sort“: 332 vs. 372 units for split and pooled recordings, respectively). I would suggest emphasizing in the manuscript that we can record from these units simultaneously in the case of pooled recordings, while in the case of split recordings we have to obtain two separate recordings to get this unit yield.

Yes, the benefit obtained in terms of simultaneous recordings was in fact understated in the manuscript. Even using the conservative attitude – in which one only considers units found in split mode recording – we were able to observe more of these cells simultaneously during electrode pooling (Line 293).

2.7. The authors should provide more detail on the probe cleaning procedure.

Now added to Methods (Line 550).

Reviewer #3 (Remarks to the Author):

The authors present a novel method to boost the yield of extracellular recordings called electrode pooling. This involves connecting the same wire to multiple switchable recording sites instead of just one electrode at a time, which is the normal practice. The authors neatly synthesize results from an analytical model, experimental data with a modified Neuropixel, and simulations to explore the merits of electrode pooling.

3.0. The paper was well written and easy to understand for the most part. The main strengths of the study are in terms of technical innovation and potential significance for switchable silicon probe makers such as the Neuropixel consortium. Despite these strengths the study was perceived to contain significant limitations which lowered overall enthusiasm. As detailed below, the main concern was lack of compelling evidence that pooling is either beneficial or practical to implement. The reviewer found a number of areas which could be improved with revisions, although it is unclear if the major concerns noted below are surmountable. The reviewer feels this study is still important but perhaps not in the way the authors intended. In the reviewer's opinion it shows that electrode pooling is unlikely to be a viable general-purpose method...but, that it may be useful in certain specialized situations involving sparse, high spike amplitude and low noise activity, and when perfect unmixing of data is not required. It is useful for the Neuropixel community to know these limitations in order to avoid false starts. Additionally, there may be

some niche applications where the benefits outweigh the tradeoffs.

Major concerns:

3.1. The biggest concern was that the paper does not provide compelling evidence that electrode pooling offers clear benefits in terms of extracellular recording yield. The authors are commended for exploring pooling using diverse approaches including a theoretical model, experiment with a modified Neuropixel, and simulated data. Unfortunately, none of those results were convincing that electrode pooling provides significant benefits in typical extracellular recording conditions.

3.1a. First, pooling only seems effective for high amplitude spikes (for example in line 356 they mention that 380 microvolts was in the 90th percentile of spike amplitudes from the Allen Institute database). Yet the same database shows that most spikes have a much lower amplitude closer to 100 microvolts.

In the revision we show that spikes can be unmixed productively even at 100 μ V amplitude, to the point where more units can be recorded in pooled mode than in split mode (Line 284, Fig 5C).

3.1b. Second, under the most ideal conditions the neuron/wire ratio was 8 (Figure 2), which would be a respectable improvement. However, this ratio drops close to ~ 1 when the conditions become less favorable and more in line with typical spike and noise amplitudes. The authors speculated in line 365 that if common noise could be reduced by a factor of 2 then the optimal pool size would increase. But throughout the paper they assumed that the only source of common noise comes from the amplifier (N_{amp}), and that biological noise N_{bio} is unique to each electrode. This seems to be an optimistic assumption, as at least part of N_{bio} may be common to many electrodes. Thus, the authors' pooled noise estimates may have been consistently underestimated, and it is unclear how a factor of 2 noise reduction would ever materialize.

See summary reply above. We have added a new figure panel to test whether N_{bio} is unique or not between the two banks in our recordings (Line 319, Fig 4F).

3.1c. Third, it seems that efforts to unmix pooled signals in Figure 5 was only moderately effective. Overall, it appears pooling may provide some modest advantages for only a select group of cells with high spike amplitude and low noise. But for the large majority of spikes measured in a recording, according to the results presented there is no unambiguous benefit to pooling, and it may even be detrimental for obtaining single cell spiking information.

We have added a figure panel that shows the benefit of pooling as a function of spike amplitude, measured experimentally. The benefit is net positive starting at 100 μ V, meaning that more units can be recorded in pooled mode than in split mode (Line 284, Fig 5C).

3.2. A related concern was that the proposed workflow for pooling described in Figure 3 seems quite complex, raising substantial concerns about the practicality of using this approach. Additionally, the four principles for choosing which electrodes to pool (lines 192-196) seem quite restrictive, further raising concerns that a large percentage of electrodes will be ineligible for pooling. To the authors' credit, they wrote a thoughtful Discussion section that outlined some of the possible opportunities for using electrode pooling in future generations of Neuropixel probes. They mentioned that pooling is likely to become more relevant as the number of recording sites greatly exceeds than the number of available wires. For this to be true, many of the problems or limitations noted in the first comment would need to be addressed, but the path to achieving that is not clear. Thus it seems uncertain if the benefit/tradeoff ratio is great enough to ever justify using this as a standard approach.

See summary reply above.

Minor comments:

3.3. Line 80: "...are summed..." Suggest changing to "averaged" as this is an important distinction as shown by equation 3.

Done.

3.4. Lines 94-95: "Sometimes a second neuron.....". Seems more appropriate to make this a general statement, eg "Sometimes multiple neurons (as many as x based on literature)....."

Done.

3.5. Figure 2: The red dots based on the experimentally derived values of alpha and beta are useful. It is suggested that authors change the dots into a shaded zone based on a range of experimentally derived values to give a better sense of what conditions are likely to be encountered. The red dot in panel 2C is based on highly optimistic alpha and beta values, and it would be useful to show a bigger range of values.

We don't have a basis for speculating about what conditions might be encountered by others. The value we plot is derived from our device measurements and a range of spike amplitudes in an online database. Readers can insert their own measured values in Eqn (10) easily enough.

3.6. Equation 4: Recommend renaming private noise as independent or

uncorrelated noise,

We mildly prefer the original term; the suggested terms would require a reference (independent of what?).

and renaming the thermal noise as N_{th} rather than N_{ele} to avoid confusion with the electronic noise.

Done.

3.7. Line 135: The authors assume that N_{bio} is independent (“private”) but it seems more conservative to assume it is common or at least partially common. By assuming N_{bio} is entirely private their beta values in equations 9 & 10 may be optimistically high, inflating the optimal M_{max} shown in Figure 2C. As noted in the next comment, since the authors have access to experimental data with pools of 2 and 3 electrodes, it should be possible to compare noise levels in split and pooled mode data.

Done in the new Fig 4F.

3.8. The noise characterization in Figures 4 and 5E could be improved. First, it would be useful to put all noise-related graphs in a single figure rather than split them between two figures.

Done.

Second, the authors are asked to show 3 histograms corresponding to the rms value of the 3 main noise types (N_{bio} , N_{amp} , $N_{thermal}$).

Done in the new Fig 4.

Third, an attempt should be made to compare how the total noise measured in vivo (N_{tot}) looks in split mode versus pooled mode using data from the modified Neuropixel probe. Ideally this might allow the authors to use Equation 6 or 7 to estimate the total amount of N_{pri} and N_{com} . In turn, this might provide insight into whether N_{bio} is predominantly “private” or “common”.

Done in the new Fig 4F.

3.9. Lines 188-190: “Finally, for every electrode one gets the private noise N_{pri} . The common noise N_{com} can be assessed ahead of time, because it is a property of the recording system.” As noted above this assumes that N_{bio} is part of the private noise. It seems more rigorous to rephrase that sentence to say that for every electrode one gets the total noise. The amplifier and thermal noise can be measured ahead of time because they are basic properties of the recording system. With this information, one can calculate N_{bio} for every electrode.

Done.

3.10. In lines 232-233, do the authors mean that the pooling function allows pooling of 2 or 3 wires, with at most 1 wire per bank? This was a bit unclear. The authors also mention that the pooling coefficient was measured at ~ 0.5 . This suggests they only used pools of 2 but never 3 wires, is that correct? If so please clarify in the text. Also, please clarify if the selection of 2 wires (one per bank) is fixed (if electrode A from bank 0 can only be pooled with electrode A from bank 1) or customizable (if electrode A from bank 0 can be pooled with any of the 384 electrodes from bank 1). Judging by some of the data and from line 304 it seems likely that the selection is fixed but it would be helpful to provide these details in Section 3.1 of the paper.

We clarified that only 2 banks were involved in these pooling experiments and that each electrode has only one associated wire (Lines 236, 294).

3.11. Figure 5: the presentation of some results in this figure was confusing and appeared to lack some crucial details. Some important information about the number of cells was delegated to Table 1 in the supplementary data, but this is crucial information that belongs in the main text and preferably within Figure 5. Additionally, panel 5B was difficult to interpret for several reasons. First, were the results obtained with cold sorting? If so the authors are requested saying so in the caption. Second, it is suggested changing the color scheme to increase contrast around cosine of 0.9. With the current color scheme it is difficult to distinguish the various red shades. Third, it is suggested somehow trying to label the 122 matched cells or grouping them together in the matrix. In panel 5C it is suggested renaming the y axis label to “Matched cells” and changing the plot style to percentage (ratio of matched cells to total original split cells) rather than absolute number. In terms of additional details it would be valuable to show a histogram of the distribution of spike amplitudes in split mode, which is useful for estimating S_{max}/S_{min} .

We adopted almost all these suggestions. The figure now includes a panel Fig 5C on the role of spike amplitude in the recovery of spikes from the pooled signal. We also revised the graphical presentation of matching in Fig 5B.

3.12. Figure 6: For the authors to make a persuasive argument that pooling is beneficial, they would need to demonstrate that pooling provides robust gains under a wide range of possible conditions. Unfortunately, in this section the exploration of various parameters that influence performance was very limited. In particular, they only explored two spike amplitudes, with the lower bound being 205 microvolts. However, 205 microvolts is still on the high end of the amplitude range, with the mean and median amplitude being closer to 100 microvolts (see Siegle biorxiv 2019, Supplementary Fig. 4). How would the simulated pooling performance look with these more typical spike amplitudes?

We now provide the spike amplitudes of real neurons that were effectively recovered from pooled recordings in Fig 5C.

Second, the authors use the Accuracy metric defined by Magland 2020, but do not report the accuracy values. It would be useful to include this information because their criterion for matched cells (accuracy greater than 0.8) seems a bit arbitrary.

We have added a figure with the distribution of accuracy values, Fig 9.

REVIEWER COMMENTS

Reviewer #1 (Remarks to the Author):

I am happy with the response given by the authors and the updates to the paper. However, authors have sometimes referred to line numbers in the manuscript that didn't match exactly to what they intended to point out (e.g., line 221 was mentioned a few times, but the information is not there). This made it harder to understand the arguments.

Reviewer #2 (Remarks to the Author):

Thanks for the detailed answers, however I still do not see a clear advantage of the method neither the well-established analytics supporting it. There are too many assumptions in the answers that may or may not prove to withstand the test of future usability. For example, there are some hypothetical mentions of the new Neuropixels 2.0 probe, with lack of analysis of real data. In my opinion, the users will either use the conventional method or the split one and will hardly attempt to use both depending on the experimental need, since switching back and forth between methods may further complicate data reproducibility.

Reviewer #3 (Remarks to the Author):

The authors made substantive improvements to the manuscript in response to reviewer comments. I especially appreciate the improved noise characterization in Fig. 4, and Fig. 5C showing that spike can be unmixed even at 100 μ V amplitude (but see my detailed comment about Fig. 5C below). Overall, the authors still tended to take an overly optimistic view of noise and signal conditions now and in the future ("lower" amplitude spikes in Fig. 6 are 205 μ V, and unclear path for reducing amplifier noise by factor of 2), so it still seems that pooling is only practical for high amplitude spikes (although clearly there are situations where high amplitude spikes are recorded on multiple electrodes and could theoretically be pooled). While the paper still has not convinced me of the merits and practicality of electrode pooling, I can appreciate that as the electrode:wire ratio on Neuropixels probes increases, some sort of intelligent electrode selection process is essential. So perhaps including a pooling option in this selection process would be a useful feature to a subset of users. Thus, this paper may stimulate discussions among the Neuropixels development team on how to design the next generation of probes.

Comment about Fig. 5C: I'm not sure I correctly interpreted what the $p=0.5$ break-even point signifies. Is this the point when 50% of spikes in pooled mode are matched to spikes in split mode? If so isn't 50% an unacceptably low percentage? I would think that $\sim 95\%$ would be a more realistic benchmark for success.

Revision 2, reply to reviewers:

Reviewer #1 (Remarks to the Author):

I am happy with the response given by the authors and the updates to the paper. However, authors have sometimes referred to line numbers in the manuscript that didn't match exactly to what they intended to point out (e.g., line 221 was mentioned a few times, but the information is not there). This made it harder to understand the arguments.

Apologies for the mixup.

Reviewer #2 (Remarks to the Author):

Thanks for the detailed answers, however I still do not see a clear advantage of the method neither the well-established analytics supporting it. There are too many assumptions in the answers that may or may not prove to withstand the test of future usability.

Certainly there is some uncertainty about future usability given that the ideal device we describe doesn't yet exist. However we believe that all the assumptions are stated clearly, and alternative sets of assumptions are explored.

For example, there are some hypothetical mentions of the new Neupixels 2.0 probe, with lack of analysis of real data.

We don't have data from Neupixels 2.0 used in this mode. The device was only announced while our paper was in review, and it is not yet available to the consumer. We hope our paper will stimulate future experiments using electrode pooling on that device.

In my opinion, the users will either use the conventional method or the split one and will hardly attempt to use both depending on the experimental need, since switching back and forth between methods may further complicate data reproducibility.

The 'split' method we describe is in fact the conventional mode of recording.

Reviewer #3 (Remarks to the Author):

The authors made substantive improvements to the manuscript in response to reviewer comments. I especially appreciate the improved noise characterization in Fig. 4, and Fig. 5C showing that spike can be unmixed even at 100 uV amplitude (but see my detailed comment about Fig. 5C below). Overall, the authors still tended to take an overly optimistic view of noise and signal conditions now and in the future ("lower" amplitude spikes in Fig. 6 are 205 uV, and unclear path for reducing amplifier noise by factor of 2), so it still seems that pooling is only practical for high amplitude spikes (although clearly there are situations where high amplitude spikes are recorded on multiple electrodes and could theoretically be pooled). While the paper still has not convinced me of the merits and practicality of electrode pooling, I can appreciate that as the electrode:wire ratio on Neupixels probes increases, some sort of intelligent electrode selection process is essential. So perhaps

including a pooling option in this selection process would be a useful feature to a subset of users. Thus, this paper may stimulate discussions among the Neuropixels development team on how to design the next generation of probes.

That would be one of our goals. Of course other manufacturers may enter this market as well.

Comment about Fig. 5C: I'm not sure I correctly interpreted what the $p=0.5$ break-even point signifies. Is this the point when 50% of spikes in pooled mode are matched to spikes in split mode? If so isn't 50% an unacceptably low percentage? I would think that ~95% would be a more realistic benchmark for success.

The fraction we quote is the opposite ratio: 50% of the spikes identified in the two combined split mode recordings can also be found in the single pooled mode recording. At this point one can record simultaneously the same number of neurons in pooled mode as in split mode. We changed the language in the text (line 288 ff) and figure caption to clarify this meaning of 'break-even'.

REVIEWERS' COMMENTS

Reviewer #3 (Remarks to the Author):

My question has been clarified and I don't have any further comments.

Response to reviewers' comments:

REVIEWERS' COMMENTS

Reviewer #3 (Remarks to the Author):

My question has been clarified and I don't have any further comments.

RESPONSE

Good.